# INDUCTIVE LINGUISTIC REASONING WITH LARGE LANGUAGE MODELS

## ABSTRACT

Evaluating large language models (LLMs) on their linguistic reasoning capabilities is an important task to understand the gaps in their skills that may surface during large-scale adoption. In this work, we investigate the abilities of such models to perform abstract multilingual reasoning through the lens of linguistic puzzles on extremely low-resource languages. As these translation tasks involve inductive and deductive reasoning from reference instances, we examine whether diverse auxiliary demonstrations can be automatically induced from seed exemplars, through analogical prompting. We employ a two-stage procedure, first generating analogical exemplars with a language model, and then applying them in-context along with provided target language exemplars. We explore various combinations of language models as analogical generators and reasoning agents, testing different model sizes and specialized multilingual LLMs. Our results on the modeLing dataset show that analogical prompting is effective in eliciting models' knowledge of language grammar similarities, boosting the performance of GPT-4o by as much as 8.1% and Llama-3.1-405B by 5.9% over chain-of-thought approaches. These gains are realized with self-generated analogical demonstrations as well as those generated by weaker multilingual models. We also report several findings about interesting phenomena which drive linguistic reasoning performance, suggesting that such puzzles are a valuable benchmark for new reasoning methods.

## 1 INTRODUCTION

As the capabilities of large language models (LLMs) continue to grow, it is necessary to develop ways of testing the boundaries of their ability to reason over a wide range of languages. In particular, adapting language models to low-resource languages is challenging due to a lack of high-quality annotated data in the target language for supervised fine-tuning. This has led to zero-shot and few-shot transfer learning approaches being more commonly employed (Zoph et al., 2016; Nguyen & Chiang, 2017; Lin et al., 2019). However, given the emergence of the in-context learning phenomenon in LLMs, we hypothesize that this behavior can be used to enable few-shot generalization to new languages at *inference time*.

In this work, we explore the task of *linguistic reasoning*, using linguistics puzzles akin to the International Linguistics Olympiad (IOL). Notably, in these puzzles, the target language is often extremely low-resource or functionally extinct (Bean et al., 2024). While prior work has largely examined the effect of vanilla in-context learning with English-target and target-English exemplars, chain-of-thought prompting, and traditional neural machine translation methods (Chi et al., 2024; Şahin et al., 2020), we believe that generating auxiliary exemplars which supplement the target language demonstrations can guide the model to more effectively learn grammar similarities over a language family. As such, we introduce an approach based on analogical prompting (Yasunaga et al., 2024), which uses strong language models to self-generate exemplars of relevant problems given the test instance and performs in-context learning conditioned on those demonstrations. In our setting, the knowledge retrieval-like nature of analogical prompting allows us to test models' parametric understanding of language families, performing inference with both the provided and induced demonstrations.

We evaluate our approach on the modeLing (Chi et al., 2024) dataset, consisting of unseen IOL-style problems. We find that strong models such as GPT-4o and Llama-3.1-405B-Instruct can identify the language family, similar languages within said family, generate exemplars in those similar

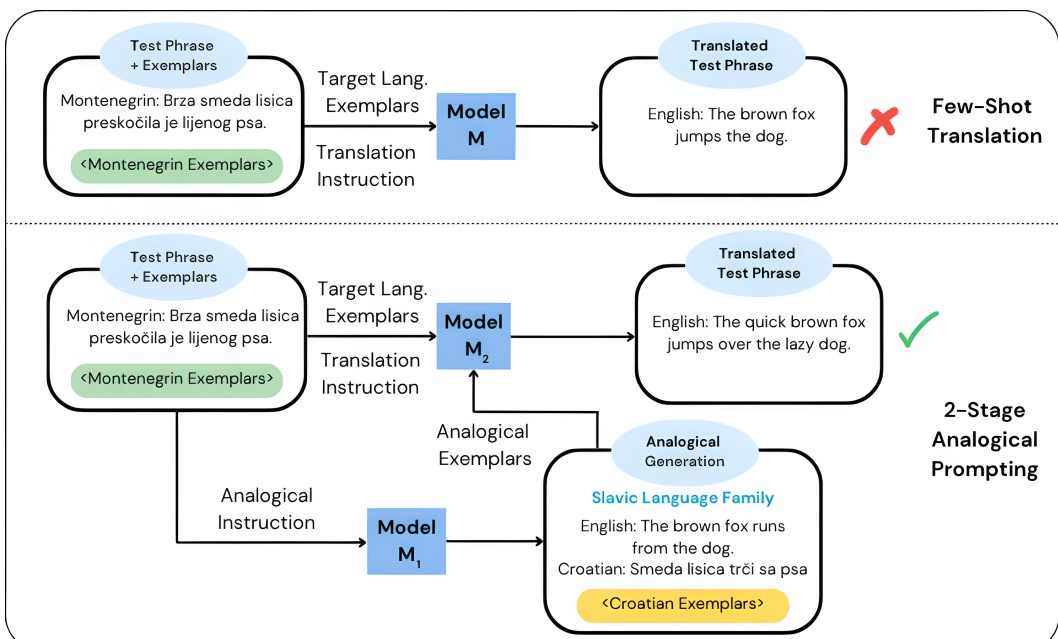

Figure 1: An illustration of our 2-stage analogical prompting approach, translating a phrase in Montenegrin to English. While prior works would solely provide exemplars translating between the source language and English and perform in-context learning, our method seeks diverse exemplars. Model $M_1$ first identifies the language family (Slavic) and higher-resource languages in the family which the model has knowledge of (Croatian), then produces exemplars in those languages. Finally, both the original and generated set of exemplars are passed with the test puzzle to model $M_2$ to perform the translation. $M_1 = M_2$ yields the self-generated analogical reasoning setting.

languages, and apply them to solve the test puzzle. Furthermore, while weak models do not benefit significantly from using strong model-generated exemplars, strong models improve from using exemplars produced by weaker yet specialized multilingual models (e.g. Aya-35B). Our findings show that the ability of the model to *deduce* and apply rules, following inductive learning from the exemplars, largely influences performance; where there is still much to be desired relative to the reasoning of human experts on this task. From our results, we suggest that the linguistic reasoning task presents a fertile ground for research on new language model reasoning methods, to uncover how the skills which drive logical thinking may be imbued to models.

## 2 ANALOGICAL PROMPTING FOR LINGUISTIC REASONING

Analogical prompting (Yasunaga et al., 2024) avoids the need for annotated exemplars by relying on a strong model to generate exemplars which are related to the test instance, but are sufficiently diverse relative to one another and the test sample. Our approach of applying analogical prompts follows the human system 2 thinking framework of slow, deliberate reasoning (Kahneman, 2011). In chain-of-thought prompting for these puzzles, the model performs in-context learning with the given exemplars, learning the rules governing the language by induction, including the meaning of particular words, and using deduction to apply these rules to the test sample. This approach is supported by prior works demonstrating the ability of LMs to learn rules and attempt to apply them (Qiu et al., 2024; Zhu et al., 2024). Furthermore, we do not have access to complete grounding sources of human-written rules governing these low-resource languages, so we must rely on the LM to identify and generate these rules itself. However, as we expect the model to have little to no prior knowledge about the target language[1], we seek to leverage other languages with similar grammar structure which the model *has* learned in order to guide the language model's reasoning process.

---

[1]We design our experiments to avoid leakage, but do not directly analyze test set contamination aside from zero-shot baselines.

We use language families as a taxonomically-grounded means of identifying similar languages as the target. The generated exemplars provide a source of reasoning support to the model, enabling it to perform inductive reasoning first in a cross-lingual manner over the diverse exemplars, and then deduce from its shared understanding. For instance, in Figure 1, given the test instance and the provided exemplars in Montenegrin, we leverage the model to (i.) identify the family of Montenegrin (Slavic Language Family), (ii.) select a few languages in the Slavic Language Family, and (iii.) generate example puzzles with their solutions in those selected languages, e.g. Croatian. Then, the provided exemplars and the generated cross-lingual demonstrations are provided to the model, to solve the given test puzzle.

In line with the view of analogical prompting as a knowledge retrieval procedure in accordance with the LM's pre-training distribution, we desire for the model to produce exemplars from languages it has learned, while probing its' understanding of language families. In (Qiu et al., 2024), models have been demonstrated to improve with more familiar exemplars (based on inclusion in the pre-training data). Furthermore, their work suggested that noisy demonstrations hurt performance; as the provided examples from the unseen target language could be considered as "noisy", we hypothesize that the generated exemplars can help to compensate.

**Exemplar Correctness.** While one would ideally prefer to have a validator which, given a set of rules for a language, can determine if they are being appropriately applied for each of the analogical exemplars, this is very challenging at scale. In the context of Linguistics Olympiad problems, only a small fraction of the population who are experts in such tasks (equivalent to achieving a high score on these contests) would be able to reliably annotate solution rationales for these extremely low-resource translation puzzles. Furthermore, the notion of correctness is ambiguous – we rely on exact match relative to an annotated "correct" response, but it is unclear if there could be more than one "correct" response which is context-specific, or if partial credit assignment could be possible. Given the models' lack of zero-shot knowledge of these languages (else, there would likely be leakage), we also cannot reliably use another language model as a validator. As a result, we leverage *all* generated exemplars by the model for inference, and assume each problem has one correct solution.

## 2.1 Linguistics Puzzles

As noted before, the focus of this work is on linguistics puzzles – in particular, translation problems from English to a low-resource language and vice versa. Such problems are also referred to in the literature as *Rosetta Stone puzzles*, and constitute one of the most frequent types of problems that appear in Linguistics Olympiad competitions (Şahin et al., 2020; Chi et al., 2024; Bean et al., 2024). These problems typically consist of a test phrase in language A along with 5-10 exemplars[2] of translation from language A to language B and vice versa, and the task is to translate the given phrase into language B. We include an example of such a problem below, from Chi et al. (2024).

---

**Example Translations from English to Rapa Nui**

English: We see you. → Rapa Nui: tikeʹa tātou koe
English: I hear you. → Rapa Nui: aroʹa au koe
English: I see you. → Rapa Nui: tikeʹa au koe
English: We hear you. → Rapa Nui: aroʹa tātou koe
English: We bite the bone. → Rapa Nui: au tātou ivi
English: We hit the bird. → Rapa Nui: puʹa tātou manu

**Translate Test Phrase**

English: The bird bites you. → Rapa Nui: au manu koe

---

## 3 Methods

We explore a number of sampling methods across various language models to assess their performance on reasoning over unknown languages.

---

[2] For more challenging problems, the model may be given as many as 20 translation exemplars.

## 3.1 BASELINES

We include the following methods as baselines for robust comparison to our method, reflecting prior work examined in linguistic reasoning (Chi et al., 2024). We explore their results in Section 4.1.

**Zero-Shot Prompting.** Given the low-resource nature of the languages that we examine, we expect zero-shot performance to be poor, or even zero, on the exact match metric. However, we include this setting for two reasons: (1.) a model which gets multiple questions correct for a given language with zero-shot prompting may be an indication of leakage, and (2.) this serves as a robust check on any additional metrics examined aside from exact match.

**Few-Shot Prompting / In-Context Learning.** As in the Linguistic Olympiad competitions, demonstrations of translation to and from the low-resource language are provided to the model, with the intention for inductive reasoning to guide the model towards identifying the set of grammar rules the language follows.

**Few-Shot Chain-of-Thought Reasoning.** Given the efficacy of chain-of-thought prompting (Wei et al., 2024; Kojima et al., 2022), we extend the few-shot evaluation setting by including prompts for the model to "think step-by-step" (Kojima et al., 2022; Yang et al., 2024a). We also include a chain-of-thought rationale exemplar for English-Spanish translation from (Chi et al., 2024), to demonstrate how step-by-step reasoning rationales should be produced, and are denoted in Section 4 as "w/ rationale".

## 3.2 ANALOGICAL PROMPTING VARIATIONS

We describe the various analogical prompting methods explored in the experiments; their results are in Section 4.2.

**Analogical Prompting on Language Families.** As noted in Section 2, we seek to use language families as a means to identify similar, auxiliary languages whose exemplars can boost the model's cross-lingual understanding. In a similar environment to the Linguistics Olympiad competition, where one does not have access to any external resources, we test the model on its latent understanding of language families and regional associations to generate further exemplars and puzzles in another language within the same language subgroup as the target language. For a target language $L$, we prompt the model to identify a few other languages (denote this list $L_{Aux}$) in the same family as $L$; then, for each language in $L_{Aux}$, generate a puzzle translating from $l$ to English, and a puzzle in the reverse direction. Then, we apply these exemplars along with the given ones for $L$ in a new instruction to the model. We term this *2-stage analogical reasoning*.

Separating the two stages of analogical prompting (generation and application) yields an opportunity to explore how different combinations of models for this approach might perform. While the above entails using the same model for both steps, we look to contrast the strength of the models used, to attempt to boost the performance of both frontier and small models.

**Inference-time Exemplar Distillation.** In our work, inference-time distillation refers to generating analogical exemplars with a strong model (e.g. GPT-4o) and applying them to a weak model (e.g. models with roughly 7-8B parameters). Our hypothesis driving this setting is: can higher quality exemplars produced by strong models enable better deductive abilities with weak models?

**Weak-to-Strong Cross-Lingual Analogies.** Specialized multilingual models such as the Aya-23 models hold promise for our linguistic reasoning analysis, as they have been fine-tuned for instruction-following across a wide range of languages. We propose using such models for generating analogical demonstrations, as they may have a stronger understanding of language families and can produce diverse exemplars, which we believe strong models may be able to deduce from.

## 3.3 EXPERIMENTAL SETUP

**Datasets.** We primarily evaluate our approaches on the modeLing dataset (Chi et al., 2024). This dataset consists of problems written by the authors and hence uninvolved in prior Linguistics

Olympiads. This benchmark was released in 2024, and we rely on its recency to be more assured that leakage is not a factor driving performance. We note that all problems examined are purely text-based; while there exist linguistics puzzles that require deduction from images, filling in diagrams, etc., the benchmark we evaluate on does not include such problems. This suggests that future work could study the performance of multimodal models on these problem types. We also evaluate on the LINGOLY dataset (Bean et al., 2024), which features 1,133 problems and expands beyond "Rosetta Stone" translation problems to include grammatical pattern-based translation, matching translation pairs, text purely in an unknown language, identifying errors in machine translation, and longer text in multiple languages. The results are included in Appendix D.

**Models.** We evaluate with the following models:

- OpenAI models: GPT-4o, GPT-4, and GPT-3.5-turbo
- Open-weight models: Llama 3.1 8B-Instruct, Llama 3.1 70B-Instruct, Llama 3.1 405B-Instruct (Dubey et al., 2024), Mixtral 8x7B-Instruct-v0.1 (Jiang et al., 2024), and Mixtral 8x22B-Instruct-v0.1
- Multilingual Instruction-tuned Models: Aya-23 8B and Aya-23 35B (henceforth referred to as Aya-8B and Aya-35B) (Aryabumi et al., 2024)

OpenAI models are inferenced with the OpenAI API, while the open-weight and multilingual instruction-tuned models are queried with the Together AI API and Apple MLX, respectively.

## 4 RESULTS

We report exact match (EM) scores for all experiments performed. ChrF2 Popović (2015), a character n-gram F-score measure, and corpus-level BLEU scores (Papineni et al., 2002) are recorded in Appendix A. We do not treat these as primary metrics as BLEU ignores word ordering nuances amidst short responses in machine translation, which is integral to measuring correctness in the puzzles we explore (Callison-Burch et al., 2006; Chi et al., 2024), and we find the ChrF scores to be noisy relative to EM scores. Smaller models with weaker instruction-following capabilities often failed to produce their output in the exact desired format specified in the prompts. To ensure that reliable exact match scores are reported while some responses may have parsing issues relative to the expected format, the authors of this work manually examined each response to confirm whether the output generated contains the target response. To enforce standardization across our evaluation procedure, this was performed for all experiments; this was not applicable for stronger models whose responses exactly followed the desired output format.

### 4.1 CHAIN-OF-THOUGHT LINGUISTIC REASONING

The results of baseline methods are in Table 1. The prompts for all experiments are included in Appendix F, and all experiments are averaged over 3 runs. For the "CoT with rationale experiment", we take the best of using 512 and 4096 max tokens (see Appendix B). For the "few shot" results, we take the best out of two different prompt settings, ablated on in Appendix C.

Our strongest baseline result is achieved with Llama-3.1-405B-Instruct producing CoT rationales, at 65.81%; in fact, this model produces the best results across all 4 baseline settings. GPT-4o remains in the high 50s, but does not exceed 60% on any single run. Among smaller models, Llama-3.1-8B-Instruct performs comparably to Aya-35B and Mixtral-8x7B-Instruct, outperforming it on some baselines, which may be attributable to a stronger and more recent base model. We also observe that GPT-4o and Llama-3.1-405B-Instruct do indeed solve a few puzzles (2 and 4 samples, respectively) in the zero-shot setting. Given the former was released before the modeLing dataset, and the latter was released just shortly after, we do not believe this to be a sign of leakage; furthermore, each correct question was from a different language.

We report a few key observations below:

**Strong models produce rationales without being instructed to.** We find that strong models such as GPT-4o and GPT-4 produce chain-of-thought stepwise rationales for responses, even in the zero-shot and few-shot settings, without including a chain-of-thought prompt or including rationales in

Table 1: Baseline experimental results using chain-of-thought methods, reporting exact match. The models have been split into three groups, corresponding to the models noted in Section 3.3. All results reported are average of 3 runs at a temperature of 0.3, to address sampling variance.

| Model | Zero-Shot | Few-Shot w/o CoT | Few-Shot w/ CoT | Few-Shot CoT w/ Rationales |
|---|---|---|---|---|
| GPT 3.5-Turbo | 0% | 25.74% | 26.10% | 38.6% |
| GPT-4 | 0% | 56.25% | 45.22% | 45.59% |
| GPT-4o | 1.10% | 59.19% | 58.82% | 55.88% |
| Llama-3.1-8B-Instruct | 0% | 22.79% | 16.91% | 23.16% |
| Llama-3.1-70B-Instruct | 0% | 45.22% | 44.49% | 42.28% |
| Llama-3.1-405B-Instruct | **1.47%** | **61.76%** | **59.19%** | **65.81%** |
| Mixtral-8x7B-Instruct | 0% | 22.43% | 22.06% | 16.18% |
| Mixtral 8x22B-Instruct | 0% | 45.59% | 43.38% | 39.71% |
| Aya-23-8B | 0% | 9.93% | 7.35% | 5.88% |
| Aya-23-35B | 0% | 23.53% | 20.59% | 14.34% |

the exemplars. This is a key reason why the few-shot without chain-of-thought setting performs the highest for both models. Furthermore, when prompted with rationale-inducing exemplars (see Appendix F), these strong models produce rule libraries from the exemplars, akin to (Zhu et al., 2024), leading to very lengthy responses; some models such as Llama-3.1 70B fall into loops of repeating the same rule many times. This necessitates the use of a higher number of max tokens to be generated, to ensure that the final answer is indeed outputted, although this makes human verification of response correctness harder due to their length; we report ablations on this in Appendix B.

**Certain models perform uncertainty-based refusal.** Some models, such as Mixtral-8x7B-Instruct and Mixtral-8x22B-Instruct respond to test instances by stating an inability to perform the desired task. This behavior especially appears in CoT with rationale exemplars; interestingly, this occurs after the implicit induction stage has been performed. For instance, with Mixtral-8x22B-Instruct, the model enumerates a set of word-level translations between the target language and English, respectively, then upon recognizing ambiguity in one of the word-level translations, it claims that solving the problem is impossible without additional information. By contrast, models such as GPT-4o instead output multiple candidate answers when it is not entirely certain. We include qualitative examples of this behavior in Appendix E. This appears to reinforce the findings of the Qiu et al. (2024) in that models are unable to reliably apply their inductively learned rules.

Our analogical reasoning method introduces an inference-time approach to boost deductive reasoning, by deliberately using their learned multilingual knowledge to guide puzzle solving.

## 4.2 Two-Stage Analogical Reasoning

To critically explore the evaluation settings introduced in Section 3.2, we select 2 frontier models – GPT-4o and Llama-3.1-405B-Instruct – which were the strongest performers in our baselines. We select 2 weaker models – Aya-35B and Llama-3.1-8B-Instruct – for the inference-time distillation and weak-to-strong prompting experiments. These models performed comparably to one another in the baselines, and allow us to contrast multilingual specialization against a generalist model with multilingual support. The experiments with Llama-3.1-8B-Instruct are included in Appendix I.

We also establish an upper bound on the performance we can attain with our approach, by a psuedo-open-book method with oracle language families. That is, for each language in the evaluation set, rather than prompting the model to implicitly infer the language family and other languages which are a member of it, we abstract away the former by providing the language family in the prompt. We suggest that a human expert with strong cross-lingual reasoning abilities would be able to deduce such relationships with similar languages, so providing language family labels eliminates one uncertainty source in the model's generations. The results of this are included in Figure 2b.

**Analogical reasoning boosts frontier models.** We find that 2-stage analogical reasoning pushes the boundaries of the performance of frontier models, relative to their best baseline results. Solely considering the self-generation setting (where the same model both generates analogical exemplars

Table 2: Pairwise results with our 2-stage analogical prompting method. The left column denotes the model generating the analogical exemplars, and the top row denotes the model applying both the generated and provided exemplars to answer the test puzzle. Note that these results address the self-generated analogical reasoning, inference-time distillation, and weak-to-strong prompting settings posed in Section 3.2.

| Generator \ Deducer | GPT-4o | Llama-3.1-405B-Instruct | Aya-23-35B |
|---|---|---|---|
| GPT-4o | 66.91% | **71.69%** | **21.32%** |
| Llama-3.1-405B-Instruct | **67.28%** | 67.65% | 20.22% |
| Aya-23-35B | 65.44% | 71.32% | 15.44% |

and applies them), GPT-4o improves 7.2% (59.19% → 66.91%), and Llama-3.1-405B-Instruct improves 1.8% (65.81% → 67.65%). We subsequently observe even stronger gains for both models as the deducer, when selecting different models as the analogical exemplar generator. In the first stage, both of these frontier models correctly identify the language family at a fairly high rate (see Appendix H), select a few languages from said family, and generate analogical puzzles for those auxiliary languages, as intended. Then, in the second stage, the model considers the tokens in the test phrase, and analyzes how each is to be translated to the target language, and combines them together in the appropriate order following the syntactical patterns observed from the given exemplars. Thus, it appears that the model uses the analogical exemplars to better induce the mappings of words in the target language to words in the source language, which it then applies to the target phrase.

**Weak analogical "supervision" improves performance.** We find that generating the analogical exemplars with Aya-35B and applying them to the test sample with Llama-3.1-405B-Instruct yields 71.32%, averaged over 3 runs; a 5.5% improvement over the best baseline for Llama-3.1-405B-Instruct (65.81% → 71.32%). We similarly find that leveraging Aya-35B-generated exemplars and applying them with GPT-4o yields a 6.2% improvement over the best GPT-4o baseline setting (59.19% → 65.44%). In the case of Llama-3.1-405B-Instruct, using Aya-generated exemplars outperforms using self-generated exemplars, by 3.7% (67.65% → 71.32%). Our findings suggest that when equipped with the right tools (analogical demonstrations) from **effective multilingual reasoners**, strong deducers can thrive.

This claim is further reinforced by the inference-time distillation results: smaller models such as Aya-35B do not benefit from the analogical exemplars, regardless of the analogical generator. At the same time, using the GPT-4o exemplars applied by Llama-3.1-405B-Instruct yields 71.69%, our strongest result across all evaluation settings. Moreover, the reverse direction (Llama-3.1-405B-Instruct exemplars applied by GPT-4o) yields an 8.09% improvement over the best GPT-4o baseline result. From these findings, we conclude that analogical exemplars generated by good multilingual reasoners do not "unlock" deductive reasoning abilities for models without them (Aya-35B); however, for strong baseline reasoners (Llama-3.1-405B, GPT-4o), better exemplars help performance.[3]

**Frontier models understand language families.** We compare model performance with and without oracle language families, finding that prompting models to infer the language family is superior. We observe that frontier models such as GPT-4o and Llama-3.1-405B, as well as specialized multilingual models like Aya-35B, have a strong parametric knowledge of language families, and do not need to rely on language family labels to identify similar languages. Furthermore, the model performing retrieval of the language family helps it to identify a few languages within the family that will help it, bootstrapping from the provided exemplars, whereas providing the language family often leads to the model listing many languages in the family and attempting to produce exemplars for all of them. We hypothesize that this is a source of noise; demonstrations beyond a certain number yield diminishing returns in performance. That is, the oracle language families setting stimulates inductive cross-lingual reasoning, but makes deductive reasoning more challenging due to having

---

[3]We note that while it would have been beneficial to acquire expert annotations on the correctness of the exemplars, this is extremely challenging given the many endangered and nearly-extinct languages present in the dataset, with only a few thousand speakers in the world.

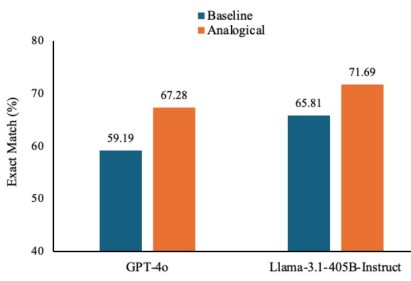 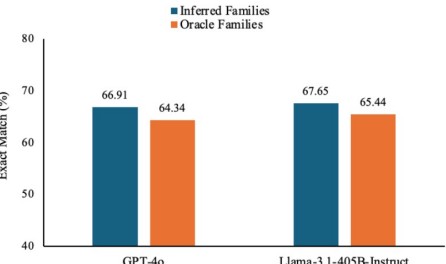

(a) Best baseline vs. best analogical      (b) Inferred families vs. oracle families

Figure 2: Figure (a) contains a comparison of the best baseline (in Table 1) with the best 2-stage analogical reasoning result (in Table 2), for our two frontier models as the deducer. We find analogical to improve GPT-4o by 8.1% and Llama-3.1-405B-Instruct by 5.9%. Figure (b) compares self-generated analogical reasoning methods, with prompt-determined language families ("inferred families") and human-annotated language family labels ("oracle families").

many exemplars. Specific examples of this behavior are included in Appendix E, and the language families table is in Appendix G. We also include further discussion on the language families identified by GPT-4o and Llama-3.1-405B-Instruct in the inferred families setting, in Appendix H, and find that they achieve a high correctness rate relative to the oracle labels.

**Language Isolates and Proxy Languages.** *Language isolates* would appear to pose particular difficulty to our models, as by definition, they do not belong to any well-defined language family. As a result, we rely solely on the models' ability to trace grammatical correspondences based on the languages it has seen in pre-training, even for our experiments with oracle language family labels. While in the baseline experiments, our models often believed that the target language is imaginary, prompting for language families leads models to note that the language is isolate. They then attempt to either follow syntactic or morphological patterns to induce a new fictitious language which is similar to the target, or select learned geographically-proximate languages. For the language of Bangime, spoken in Mali, the model either retrieves languages from families in the same geographical region, such as Dogon, or creates a new language (e.g. "Xangime") for which it generates analogical exemplars (see Appendix E). Analyzing at the instance level, this improves the correctness on the Bangime puzzles from 27.8% to 50% for GPT-4o in the self-generated setting.

In summary, our results suggest that the ability of the model to deduce by leveraging the given and analogically-generated exemplars is the key performance driver. This is lent credence by the efficacy of weak-to-strong prompting (i.e. relying on the exemplars of Aya-35b), while the performance of inference-time distillation remains roughly similar. Thus, we posit that the "strength" of a linguistic reasoning agent can be decomposed along two lenses, corresponding to our two stages: (1.) **generating analogical exemplars by language identification and multilingual reasoning**, and (2.) **deducing from hypotheses in complex evaluation settings**.

## 5    RELATED WORK

### 5.1    LARGE LANGUAGE MODEL REASONING.

**Few-shot Chain-of-Thought Reasoning.** In-context learning has emerged as an exciting phenomenon in language models, enabling them to learn from few-shot demonstrations at inference-time to generalize to various tasks (Brown et al., 2020; Wei et al., 2022). At the same time, the chain-of-thought (CoT) reasoning method guides language models to think systematically through a problem, in a step-by-step manner (Wei et al., 2024; Kojima et al., 2022). In particular, applying chain-of-thought prompting (zero-shot or few-shot) with the goal to induce rationales yields explanations of why each step was performed, along with performance and faithfulness improvements (Nye et al., 2021; Lampinen et al., 2022). Various similar approaches (Yao et al., 2023; Wang et al., 2023; Besta et al., 2024) have been proposed to sample more diverse generations from models, leveraging test-time compute to improve reasoning performance; we believe such methods make for interesting lines of future research for the linguistic reasoning task.

**Inductive Reasoning in LLMs.** Inductive and deductive reasoning skills in language models have often been studied in the context of logical or abstract reasoning problems. Much of this prior work on inductive reasoning with language models studies evaluation settings with more clearly defined rules to be inductively learned and then applied; these works suggest gaps relative to the human intelligence in performing both inductive and deductive reasoning (Xu et al., 2024; Gendron et al., 2024; Yang et al., 2024c). In particular, Yang et al. (2024c) notes the need for more challenging tasks in inductive reasoning to better assess the boundaries of LM capabilities, such as hypothesis generation and pattern induction. Works such as Tang et al. (2023) demonstrate that models struggle to create rules by induction when the semantics of the exemplars do not follow in a commonsense manner – in our work, generating analogical exemplars similar to the models pre-training data may steer the model towards a relative "commonsense" representation of the rules underlying the exemplars.

Several works dive into the realm of hypothesis search, determining the ability of LMs to pose hypotheses about the problem (e.g. rules which exemplars follow) before seeking to deductively apply them (Zhu et al., 2024; Qiu et al., 2024; Wang et al., 2024). Zhu et al. (2024) propose hypotheses-to-theories (HtT), which learns a rule library from an induction stage, and then applies it by a deduction stage; this multi-stage method is similar to our analogical approach, although we still perform both induction and deduction together after analogical generation. Furthermore, their rule library depends on verification – this is not possible in the linguistic reasoning task due to the lack of a reliable feedback source to judge responses, aside from expert humans. As discussed earlier in our work, Qiu et al. (2024) demonstrates that models can propose rules well, but cannot consistently apply them. Wang et al. (2024) proposes Hypothesis Search, a method which proposes hypotheses, implements a subset of them as Python programs, and applies them to training samples to verify their correctness.

**Exemplar Generation and Automated Reasoning.** Analogical prompting (Yasunaga et al., 2024) has been demonstrated to be an effective inference-time method to produce diverse, task-conditioned exemplars, improving in-context learning. As noted above, this effectively serves as a knowledge retrieval method which retrieves exemplars similar to (or directly from) the pre-training distribution which the model has seen; RECITE (Sun et al., 2023) similarly retrieves passages directly from the model's memory. Methods such as SG-ICL (Kim et al., 2022) and Auto-ICL (Yang et al., 2024b) also self-generate in-context exemplars in a similar manner as analogical prompting.

## 5.2 MULTILINGUAL REASONING.

Multilingual reasoning in LMs for low-resource languages poses a unique challenge, as the pretraining corpora and supervised fine-tuning datasets for many models are largely concentrated on a few high-resource languages. XLT (Huang et al., 2023) introduces a prompt template which translates problems in other languages to English and solves the problem with chain-of-thought in English. Qin et al. (2023) aligns each step in the chain-of-thought between the source language and English explanations, then solves the problem given this alignment; they also apply self-consistency with cross-lingual alignments with a set of pre-specified target languages. Li et al. (2024) trains on code data with multilingual comments, while using multilingual code prompts at inference time with symbolic function API calls as a structured way to solve the reasoning problem.

**Linguistic Reasoning Benchmarks.** The PuzzLing Machines dataset (Şahin et al., 2020) first introduced a set of Linguistics Olympiad problems to study the ability of language models to learn from a small amount of data; they apply RoBERTa-based neural machine translation methods, but demonstrate a vast gap (attaining less than 4% exact match performance). With concerns of potential leakage given the vast web scraping performed in procuring pre-training tokens for language model training, modeLing (Chi et al., 2024) introduced a new set of hand-written Linguistics Olympiad problems, demonstrating the performance of current models with CoT methods. The LINGOLY (Bean et al., 2024) dataset presents problems from the UK Linguistics Olympiad competition, and studies zero-shot and few-shot performance of current models categorized by question type.

## 6 DISCUSSION

We propose applying analogical prompting as a test of inductive reasoning from diverse exemplars for challenging linguistic puzzles. Our results encouragingly suggest that despite frontier models struggling with deductive reasoning, in line with the findings of (Qiu et al., 2024), they can indeed

follow grammar rule similarities to generate analogical demonstrations and attempt to apply them adeptly. This yields improved performance in self-generated analogical prompting with GPT-4o and Llama-3.1-405B, as well as weak-to-strong prompting for those models employing Aya-35B-generated demonstrations. We also show that Llama-3.1-405B-Instruct is the best model for linguistic reasoning at present, becoming the first model to achieve over 70% on the modeLing benchmark by way of our 2-stage analogical reasoning approach. This could be attributed to the vast scale of multilingual pre-training data (15T tokens), as well as multilingual supervised fine-tuning and dialogue data seen in adaptation. The multilingual prowess of the Llama-3.1 models is evidenced by its strong performance on the MGSM and multilingual MMLU benchmarks (Dubey et al., 2024).

Furthermore, the ability of smaller and specialized multilingual models (Aya) to generate coherent analogical exemplars, which improve frontier models over their own self-generated exemplars, is promising towards developing widely-available multilingual reasoners. We find that the improvements observed can be attributed to the auxiliary exemplars generated, which are in turn due to the model's understanding of language families and grammar rules from the pre-training data or its multilingual adaptation. The errors made by current models due to an inability to apply diverse and complex exemplars suggest that the linguistic reasoning task is an exciting and challenging evaluation setting for LM reasoning at large. That is, seeking to emulate human reasoning, where deduction involves a clear application of recognized patterns, provides a ripe space for future work.

The interesting phenomenon examined with language isolates also provides a glimpse of model capabilities to follow grammatical similarities, rather than relying on knowledge retrieval of language families. That is, the multilingual language understanding abilities of frontier models expand beyond typological knowledge, going so far as to create proxy fictitious languages which enable it to solve the problem correctly. We suggest that future efforts in multilingual adaptation be placed in identifying techniques to guide languages models to support typologically unique languages.

Research at the intersection of machine translation and reasoning in the development of the latest foundation models is important from a societal perspective. With large language models being adopted widely, the need for multilingual capabilities and rapid adaptation grows, and our work proposes an effective method by which this can be performed at test-time. Notably, we have demonstrated evidence that models follow language similarities – given the massive number of languages and dialects present worldwide, this could help guide humans to learn dying languages, thus keeping their tradition alive, while doing so in a scalable manner. We hope that these findings can inspire future models releases to include evaluation on challenging multilingual tasks such as these puzzles, and research on reasoning can explore the multilingual setting further in depth.

**Limitations** We note that the reliance on exact match scoring as our primary signal of performance is not ideal, as it is a binary indicator. We have sought to examine other metrics which correspond to "partial credit" such as ChrF2 and BLEU; however, there are flaws in these methods as well. A stronger *human understanding* of the rules which these extremely low-resource languages follow could guide us to better metrics, especially capturing semantic meaning and word ordering inversions, where appropriate. For instance, some languages might retain the same meaning while inverting the word order – exact match is sensitive to this, and while ChrF2 and BLEU are not, we should *only* be insensitive to ordering for languages which follow this property. We also recognized that the IOL 2024 problems could not be used as a benchmark with our method, as they require multimodality – our method only analyzes unimodal text problems. Another limitation of our work is that we do not have a reliable means of verifying the correctness of analogical exemplars, nor contrasting the quality of exemplars generated across models to determine the best analogical generator model. An expert annotator who could identify where a mistake was made in the model's reasoning process also would have been helpful to yield further insights into the fallacies of current models' linguistic reasoning. Nonetheless, our most effective deducer models are able to leverage exemplars generated by models of various sizes for improved linguistic reasoning.

**Reproducibility Statement.** We include all prompts used for generating our baseline experimental results, and all analogical prompting methods, in Appendix F. We have also broken down the two stages of our analogical reasoning method for clarity on how the method should be applied with two separator models (e.g. weak-to-strong prompting, inference-time exemplar distillation). We evaluate our work on the modeLing dataset, which is publicly available. We have included details of the platforms through which the models we evaluate have been queried (OpenAI API, TogetherAI API, Apple MLX), along with the list of models studied. Lastly, we will release our code publicly.

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

# A RESULTS WITH CHRF AND BLEU METRICS

While our primary results are included in Section 4 with exact match scoring, we also include the ChrF2 and BLEU scores for those experiments. Although exact match is helpful for assessing performance on absolute terms, character and word-level metrics can help in determining partial progress. While the challenges of using BLEU are discussed in Section 4, we include the corpus-level scores as it is a commonly-employed metric in machine translation settings. We use the ChrF2 score (Popović, 2015) as implemented in SACREBLEU (Post, 2018); this metric doubles the precision value in the denominator of the F-score, placing more value on the recall. The inclusion of a character-level metric is useful for robustness to morphologically rich languages in our low-resource setting.

## A.1 CHRF2 SCORES FOR BASELINE EXPERIMENTS

Table 3: Baseline experiments as reported in Table 1, but with the ChrF2 metric instead.

| Model | Zero-Shot | Few-Shot w/o CoT | Few-Shot w/ CoT | Few-Shot CoT w/ Rationales |
|---|---|---|---|---|
| GPT 3.5-Turbo | 4.37 | 30.61 | 12.93 | 37.50 |
| GPT-4 | 32.61 | 38.46 | 35.71 | 40.54 |
| GPT-4o | 37.50 | 39.47 | 40.54 | 40.54 |
| Llama-3.1-8B-Instruct | 0.25 | 40.54 | **48.39** | **45.45** |
| Llama-3.1-70B-Instruct | 38.46 | 34.09 | 38.46 | 41.67 |
| Llama-3.1-405B-Instruct | 27.27 | 38.46 | 38.46 | 38.46 |
| Mixtral-8x7B-Instruct | 39.47 | 4.10 | 1.49 | 12.30 |
| Mixtral 8x22B-Instruct | **42.86** | 38.46 | 2.42 | 34.88 |
| Aya-23-8B | 21.13 | 39.47 | 30 | 41.67 |
| Aya-23-35B | 27.27 | **46.88** | 46.88 | **45.45** |

These results seem to suggest that while they do not perform as well as the frontier models on exact match, Llama-3.1-8B-Instruct and Aya-35B attain high ChrF2 scores, due to being close to the target translation, but e.g. making a few character insertions or deletions, or word order changes. To that effect, ChrF2 serves as a useful measure of "partial credit".

## A.2 BLEU SCORES FOR BASELINE EXPERIMENTS

Table 4: Baseline experiments as reported in Table 1, with corpus-level BLEU scores.

| Model | Zero-Shot | Few-Shot w/o CoT | Few-Shot w/ CoT | Few-Shot CoT w/ Rationales |
|---|---|---|---|---|
| GPT 3.5-Turbo | 0.06 | 5.33 | 14.65 | 19.96 |
| GPT-4 | 0.52 | 40.07 | 16.70 | 6.14 |
| GPT-4o | **0.75** | **50.53** | **34.76** | **36.33** |
| Llama-3.1-8B-Instruct | 0.02 | 0.54 | 0.09 | 0.06 |
| Llama-3.1-70B-Instruct | 0.47 | 0.65 | 0.57 | 0.36 |
| Llama-3.1-405B-Instruct | 0.19 | 3.34 | 1.22 | 6.28 |
| Mixtral-8x7B-Instruct | 0.04 | 0.52 | 0.32 | 0.31 |
| Mixtral 8x22B-Instruct | 0.09 | 11.36 | 3.84 | 7.45 |
| Aya-23-8B | 0.04 | 4.54 | 4.24 | 5.88 |
| Aya-23-35B | 0.12 | 11.37 | 11.55 | 0.58 |

We find that BLEU scores are highest for GPT-4o. However, this is a somewhat noisy signal, as Llama-3.1-405B attains the highest exact match performance, but very low corpus-level BLEU scores, below several models which it outperforms on the stricter (EM) metric.

## A.3 CHRF2 SCORES FOR ANALOGICAL REASONING EXPERIMENTS

Table 5: Analogical reasoning experiments as reported in Table 2, with ChRF2 scores.

| Generator \ Deducer | GPT-4o | Llama-3.1-405B-Instruct | Aya-23-35B |
|---|---|---|---|
| GPT-4o | 40.54 | 38.46 | 46.88 |
| Llama-3.1-405B-Instruct | 40.54 | 42.86 | 46.88 |
| Aya-23-35B | 38.46 | 32.86 | 46.88 |

## A.4 BLEU SCORES FOR ANALOGICAL REASONING EXPERIMENTS

Table 6: Analogical reasoning experiments as reported in Table 2, with corpus-level BLEU scores.

| Generator \ Deducer | GPT-4o | Llama-3.1-405B-Instruct | Aya-23-35B |
|---|---|---|---|
| GPT-4o | 39.50 | 6.95 | 3.66 |
| Llama-3.1-405B-Instruct | 41.76 | 2.35 | 2.82 |
| Aya-23-35B | 30.27 | 3.11 | 3.81 |

## B ABLATIONS ON MAX TOKEN LENGTHS FOR RATIONALE GENERATION

For the chain-of-thought baseline where English-Spanish translation with rationales is provided (from Chi et al. (2024)), we observe that frontier models produce verbose outputs. These outputs include explaining the meaning of each word in the exemplars for the target language (inductive learning), before applying them to the test sample. We find that including a max token length of 4096 as opposed to 512 yields vastly different results.

Table 7: A comparison of values of max tokens to generate, 512 against 4096.

| Model | 512 Max Tokens | 4096 Max Tokens |
|---|---|---|
| GPT 3.5-Turbo | 30.51% | 38.60% |
| GPT-4 | 41.91% | 45.59% |
| GPT-4o | 55.51% | 55.88% |
| Llama-3.1-8B-Instruct | 19.85% | 23.16% |
| Llama-3.1-70B-Instruct | 42.28% | 1.1% |
| Llama-3.1-405B-Instruct | 37.87% | 65.81% |
| Mixtral-8x7B-Instruct | 16.18% | 11.76% |
| Mixtral 8x22B-Instruct | 30.88% | 39.71% |

In particular, we find that Llama-3.1-405B-Instruct, Mixtral-8x22B-Instruct-v0.1, and GPT-3.5-Turbo improve significantly, by over 8%. Notably, Llama-3.1-405B-Instruct with the ability to generate up to 4096 tokens yields our strongest baseline result of 65.81%. Conversely, Llama-3.1-70B-Instruct surprisingly drops to 1.1%, performing almost as poorly as the zero-shot baseline. Upon manual inspection, we find this to be due to entering loops of repeating the same rationale step until the max token limit is reached.

## C  FEW-SHOT PROMPT ABLATIONS

We also include the results with the provided few-shot exemplars, while using two different instructions. The "zero-shot prompts" are the system prompt and instruction used for zero-shot evaluation, where no reference is made to the existence of few-shot exemplars. The few-shot prompt used is a close adaptation of that used in Chi et al. (2024). Surprisingly, we find that this makes a slight, yet noticeable difference in results. The prompts used can be found in Appendix F.

Table 8: Comparison between two different few-shot prompting scenarios; the first involves providing the exemplars to the model, but making no mention of them in the instruction. The later also provides the exemplar, but instructs the model to only use those to solve the problem.

| Model | "Zero-Shot" Prompts | 4096 "Few-Shot" Prompts |
|---|---|---|
| GPT 3.5-Turbo | 25.74% | 12.50% |
| GPT-4 | 56.25% | 53.68% |
| GPT-4o | 58.09% | 59.19% |
| Llama-3.1-8B-Instruct | 21.32% | 22.79% |
| Llama-3.1-70B-Instruct | 42.65% | 45.22% |
| Llama-3.1-405B-Instruct | 60.29% | 61.76% |
| Mixtral-8x7B-Instruct | 11.76% | 22.43% |
| Mixtral 8x22B-Instruct | 45.59% | 44.49% |

Notably, GPT-3.5-Turbo and GPT-4 perform better with the "zero-shot prompts"; we believe this to be attributable to the few-shot prompt specifying to solve the puzzle *only* using the in-context exemplars. This perhaps could be limiting the model from drawing from its knowledge base to solve the problem. At the same time, Mixtral-8x7B performs much better with the few-shot prompts.

## D  LINGOLY DATASET RESULTS

To further the generalizability of our findings, we also evaluate our 2-stage analogical prompting method on the LINGOLY dataset (Bean et al., 2024). This dataset includes 1,133 subquestions across 90 languages, derived from the UK Linguistics Olympiad (UKLO), and features several problem types beyond the Rosetta Stone category which constitutes the primary focus of our work (although Rosetta Stone problems form 46% of the dataset). These categories include Pattern (translation based on grammatical patterns), Match-up (matching translation pairs), Monolingual (text purely in an unknown language), Computational (identifying errors in machine translation), and Text (longer text in multiple, often higher-resource languages). The difficulty levels vary from Breakthrough (easiest, for newcomers of the UKLO), Foundation, Intermediate, Advanced, and Round 2 (hardest, invitational qualifier for the IOL). As such, applying our approach with this dataset serves as a valuable test of the transferability of this method across datasets and cross-lingual tasks.

Table 9: Baseline results with GPT-4o, as reported in LINGOLY (Bean et al., 2024), on exact match.

| | Computational | Text | Monolingual | Match-up | Pattern | Rosetta |
|---|---|---|---|---|---|---|
| Breakthrough | | 100% | | | 47% | 79% |
| Foundation | 0% | | | 100% | 67% | 62% |
| Intermediate | | | | | 58% | 34% |
| Advanced | | | 0% | 33% | 53% | 26% |
| Round 2 | | | 0% | 30% | 27% | 12% |

We report the results for GPT-4o in the self-generated analogical prompting setting, in a tabular format, as well as in pictorial representations in Figures 3 and 4 in the bubble plot style of Bean et al. (2024). We demonstrate the performance (exact match scores) of the model in each combination (difficulty level and question type), as well as the improvements (denoted $\Delta_{Baseline}$) over

Table 10: Results with Two-Stage Analogical Prompting (Ours) with GPT-4o on exact match.

| | Computational | Text | Monolingual | Match-up | Pattern | Rosetta |
|---|---|---|---|---|---|---|
| Breakthrough | | 100% | | | 80% | 86% |
| Foundation | 0% | | | 100% | 69% | 80% |
| Intermediate | | | | | 83% | 64% |
| Advanced | | | 19% | 50% | 73% | 51% |
| Round 2 | | | 14% | 42% | 49% | 41% |

Table 11: $\Delta_{Baseline}$, the improvement yielded by our Two-Stage Analogical Prompting method over the baseline results.

| | Computational | Text | Monolingual | Match-up | Pattern | Rosetta |
|---|---|---|---|---|---|---|
| Breakthrough | | 0% | | | +33% | +7% |
| Foundation | 0% | | | 0% | +2% | +18% |
| Intermediate | | | | | +25% | +30% |
| Advanced | | | +19% | +17% | +20% | +25% |
| Round 2 | | | +14% | +12% | +22% | +29% |

the baseline results reported in Bean et al. (2024). We followed the same evaluation procedure as we did with modeLing, handling parsing issues accordingly for reliable exact match scoring. Note that all categories for which the table is empty are those for which no problem of that type exists in the dataset at present (or rather, there has not been such a problem in the recent history of the UK Linguistics Olympiad, from which the dataset was curated).

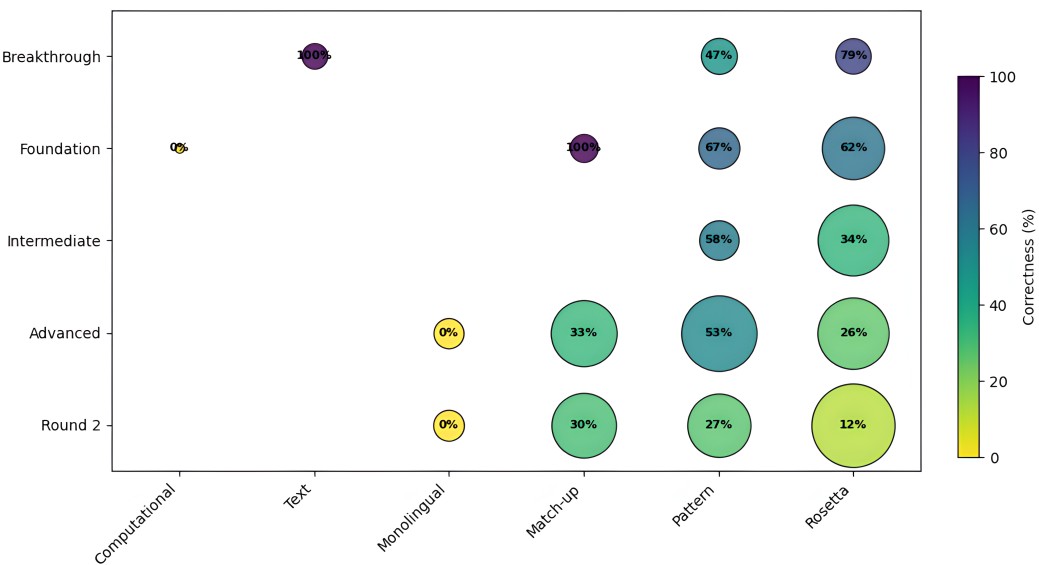

Figure 3: Baseline Results with GPT-4o on LINGOLY. The size of the bubbles correspond to the number of subquestions of that type present in the dataset.

We find that our results significantly outperform the baseline by a sizable amount across all difficulty levels, and across all tasks. Moreover, the results outperform the Claude-3 Opus state-of-the-art scores reported in the LINGOLY paper on every single setting, with the exception of the Breakthrough Rosetta Stone (easiest problems). Specifically, we find that our 2-stage analogical prompting method enables GPT-4o to solve questions of the monolingual type which it could not

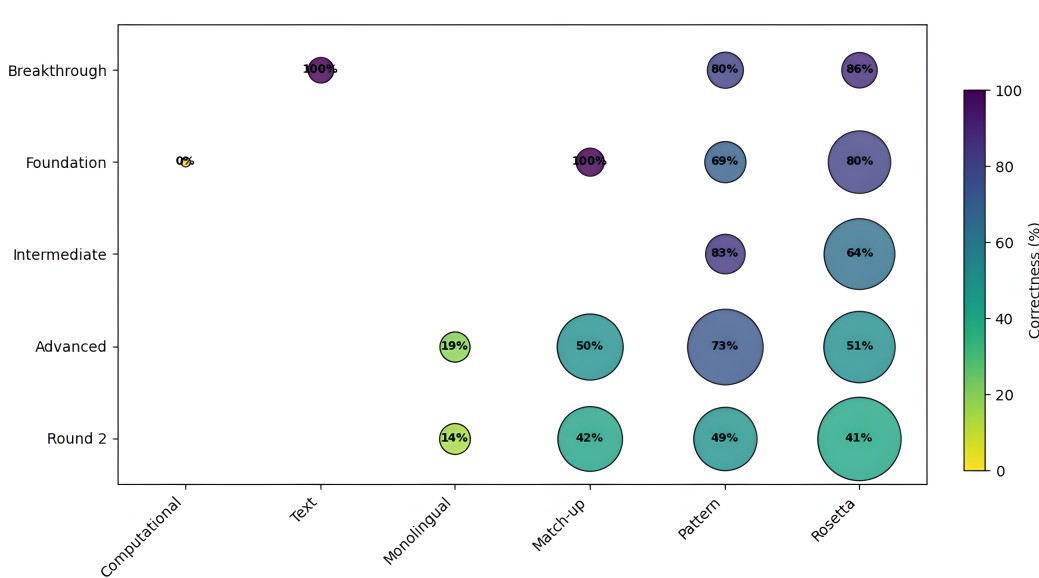

Figure 4: Two-Stage Analogical Prompting (Ours) Results with GPT-4o on LINGOLY. The size of the bubbles correspond to the number of subquestions of that type present in the dataset.

before (0% → 19% and 14%); furthermore, the correctness rates jump considerably for some of the hardest categories over the baseline (1.81x in Round 2 Pattern, 1.96x in Advanced Rosetta Stone, and 3.42x in Round 2 Rosetta Stone). It is especially worth noting that the Round 2 Rosetta Stone results corroborate with our findings on modeLing as reported in our paper. These findings suggest that our method generalizes across both datasets and question types.

# E QUALITATIVE EXAMPLES

## E.1 ANALOGICAL DEMONSTRATIONS

Analogical exemplars generated by GPT-4o for target language Ngadha:

> "Based on the provided translations, it appears that Ngadha is a language that belongs to the Austronesian family, specifically within the Central-Eastern Malayo-Polynesian group. Other languages in this subgroup include those from the Flores-Lembata languages, such as Rembong and Lamaholot.are some example puzzles translating from Rembong and Lamaholot to English, and vice versa. Translating from Rembong to English: ua gina : big house : ga'e buin : small flower : kafi gala : black chair : watu lin : long stone : regu boku : white cloud : poa mina : green forest Translating from English to Rembong: big river : rehu gina : small bird : ki'i buin : black mask : meke gala : long road : jalan lin : white sky : ara boku : green grass : lea mina Translating from Lamaholot to English: mawar gete : big rose : sarik wena : small rock : moto wulan : black moon : jalan rake : long street : lopo puti : white hut: nara hoban : red boat Translating from English to Lamaholot: big tree : po'o gete : small river : wae wena : black cloud : lero wulan : long valley : tana rake : white mountain : weling puti : red fruit : bua hoban"

GPT-4o generated analogical exemplars for target language:

> "**[Example Puzzle: Quechua to English]**: Ŏ0f1awi: eye : hamuy: come : wasi: house : sacha: forest : yacu: water **[Test Puzzle: Quechua to English]**: llapa: **[all]** : pacha: **[time]** : chay: **[that]** : puŎ0f1uy: **[sleep]** : qhari: **[man]****[Example Puzzle:

Nahuatl to English]**: atl: water : calli: house : mixtli: cloud : teotl: god : cuauhtli: eagle **[Test Puzzle: Nahuatl to English]**: tlitl: **[fire]** : yollotl: **[heart]** : in: **[the]** : michin: **[fish]** : huitzilin: **[hummingbird]**"

Analogical exemplars generated by Aya-23-35B for target language Kalam (an example of generating new languages which it believes to be similar as Kalam):

"Here are some puzzles for translating from and to languages in the same family as Kalam: **Puzzle 1: Translating from Languge X to English**: fey: fly: fey-gaw: (fly) away: gaw: go: gaw-fey: (go) away: nack: take: nack-fey: take (and) fly: nack-gaw: take (and) go: nack-fey-gaw: take, fly, and go**Puzzle 2: Translating from English to Language Y**: walk: yack: walk quickly: yack-yack: run: yack-fey: run away: yack-fey-gaw: take and walk: nack-yack: take and fly: nack-fey: take, walk, and fly: nack-yack-fey**Puzzle 3: Translating from English to Language Z**: eat: ze: eat quickly: ze-ze: drink: ze-fey: drink water: ze-fey-gaw: take and eat: nack-ze: take, eat, and drink: nack-ze-ze-feypuzzles help establish the relationships between the languages and English, with each puzzle focusing on a specific theme (going/flying, eating/drinking, and taking) to illustrate the translation patterns. note that the languages X, Y, and Z are fictional and used for illustrative purposes only."

Analogical exemplars generated by Aya-23-35B for target language Bangime:

"Here are some puzzles translating from and to languages in the same family as Bangime: **Puzzle 1: Translating from Mande languages to English**- Manding: M̈i na soo.¨: Ï see you.¨- Manding: T̈a na firi.¨: Ẅe hear you.¨- Manding: T̈a na biri.¨: Ẅe bite the bone.¨**Puzzle 2: Translating from English to Mande languages**- English: T̈he dog drinks.¨: D̈ugul̆00fc na kur̆00fc.¨- English: T̈he fish sleeps.¨: P̈ishul̆00fc na sara.¨**Puzzle 3: Translating from Niger-Congo languages to English**- Igbo: M̈ uwe.¨: Ï see you.¨- Igbo: M̈ nwunye.¨: Ẅe hear you.¨- Igbo: M̈ gbuo.¨: Ẅe bite the bone.¨**Puzzle 4: Translating from English to Niger-Congo languages**- English: Ï am hitting.¨: M̈ uwu.¨- English: T̈he person drinks.¨: Öme na kur̆00fc.¨- English: T̈he bone sleeps.¨: Ëji na sara.¨puzzles showcase the translation relationships between Bangime and other languages in the same language family, helping to establish patterns and rules for accurate translation."

### E.2 Uncertainty-Based Refusal

Mixtral-8x22B-Instruct-v0.1 expressing an inability to answer the question, despite producing a "rule library" based on the given exemplars. This is due to the incompleteness of the library in understanding what the phrase "mise" means, from the exemplars.

Based on the given examples, we can infer the following:

1. "eni" means "we" 2. "moni" means "see" 3. "syiene" means "hear" 4. "wo" means "you" 5. "fyani" means "fly" 6. "dhi" means "eat" 7. "ubhwo" means "bird" 8. "mise" is not given a direct translation in the examples, but we can infer that it is a noun because it follows the pattern of other nouns in the sentences.

Given the sentence "eni mise", we can break it down as follows:

1. "eni" means "we" 2. "mise" is a noun, but we don't have a direct translation for it.

Therefore, the sentence "eni mise" translates to "We [mise]". Without more context or examples, we cannot provide a more specific translation for "mise".

## F    Prompts Used in Experiments

### F.1    Zero-Shot Prompts

#### F.1.1    System Prompt

'You are an experienced linguist with background in a wide variety of languages, and translating them to and from English. You have been asked to translate a series of phrases from a language to English, or from English to that language. You have never seen this language before, but you are confident in your ability to translate the phrases accurately.'

### F.1.2 INSTRUCTION

'This is a translation puzzle. Here is a phrase in Language (a never-seen-before foreign language) or in English. If the test phrase is in English, your task is to translate it into Language. If the test phrase is in Language, your task is to translate it into English. When you are done with your answer, provide your outputs in the format of **[your answer]**.'

## F.2 FEW-SHOT AND ANALOGICAL REASONING SYSTEM PROMPT

'You are an experienced linguist with background in a wide variety of languages, and translating them to and from English. You have been asked to translate a series of phrases from a language to English, or from English to that language. You have never seen this language before, but you have been given a few examples of phrases in the language and their English translations to help you. You are confident in your ability to translate the phrases accurately.'

## F.3 FEW-SHOT, NO CHAIN-OF-THOUGHT

'This is a translation puzzle. Below are example phrases in Language (a never-seen-before foreign language) as well as their English translations. Some test phrases follow them. If the test phrase is in English, translate it to Language; if the test phrase is in Language, then translate it to English. Your task is to look closely at the example phrases and use only the information from them to translate the test phrases. When you are done with your answer, provide your outputs in the format of **[your answer]**.'

## F.4 FEW-SHOT WITH CHAIN-OF-THOUGHT, NO RATIONALE

'This is a translation puzzle. Below are example phrases in Language (a never-seen-before foreign language) as well as their English translations. Some test phrases follow them. Your task is to look closely at the example phrases and use only the information from them to translate the test phrases. If the test phrase is in English, translate it to Language; if the test phrase is in Language, then translate it to English. Take a deep breath and work on this problem step-by-step in a logical way, using careful analytical reasoning to get the correct result. When you are done with your answer, provide your outputs in the format of **[your answer]**.'

## F.5 FEW-SHOT CHAIN-OF-THOUGHT WITH RATIONALE PROMPT

'This is a translation puzzle. In a moment, you will use logic and analytical reasoning to translate from a never-seen-before language (Language) to English. If the test phrase is in English, translate it to Language; if the test phrase is in Language, then translate it to English. As a training example, here are some expressions in Spanish and their translations in English.
1. Spanish: ventana roja English: red window
2. Spanish: ventana azul English: blue window
3. Spanish: manzana azul English: blue apple
Using the above examples, translate the following. Spanish: manzana roja
EXPLANATION: The first step we notice is that the word "ventana" must mean window because (1) the word "ventana" appears twice between sentences 1 and 2, and (2) the only word that appears twice in the English translation is "window." Next, we infer that "roja" must be "red" and "azul" must be "blue" by process of elimination. Next, we guess that in Spanish, the noun precedes the adjective because "ventana" comes before "roja" and "azul." Therefore, the

noun in sentence 3 ("apple") must correspond to the word preceding the adjective ("manzana") in the Spanish translations. Putting this together, "manzana roja" must mean "red apple" in English.
ANSWER: English: red apple.
Now, given the following test phrase, please translate it. Take a deep breath and work on this problem step-by-step in a logical way, using careful analytical reasoning to get the correct result. When you are done with your answer, provide your outputs in the format of **[your answer]**.'

### F.6 ONE-STAGE ANALOGICAL PROMPTING

"This is a translation puzzle. In a moment, you will use logic and analytical reasoning to translate from a never-seen-before language (Language) to English. Given a few example puzzles translating from Language to English (or English to Language), generate 3 similar puzzles translating other languages in the same family as Language to English, and 3 similar puzzles translating from English to those languages in the same family as Language. The puzzles that you generate should be distinct from one another, the example puzzles, and the test puzzle. They also should be from a diverse set of languages within the same language family as the test puzzle. Your task is to look closely at the example puzzles and the puzzles that you have generated in order to solve the test puzzle. Take a deep breath and work on this problem step-by-step in a logical way, using careful analytical reasoning to get the correct result. When you are done with your answer, provide your outputs in the format of **[your answer]**."

### F.7 TWO-STAGE ANALOGICAL PROMPTING

#### F.7.1 ANALOGICAL EXEMPLAR GENERATION PROMPT, INFERRED LANGUAGE FAMILIES

"Given a few example puzzles translating from {name} to English (or English to {name}), identify few other languages in the same family as {name}, generate a puzzle similar to translating other languages in the same family as {name} to English, and another puzzle translating from English to those languages in the same family as {name}. The puzzles that you generate should be distinct from one another than the example puzzles, and the test puzzle but should help establish the relationships for translation between {name} and English. They also should be from a diverse set of languages within the same language family as the test puzzle. Provide your outputs in the format of **[your answer]**."

#### F.7.2 ANALOGICAL EXEMPLAR GENERATION PROMPT, ORACLE LANGUAGE FAMILIES

"Given a few example puzzles translating from name to English (or English to {name}), identify few other languages in the {lang_family} family, generate a puzzle similar to translating other languages in the same family as {name} to English, and another puzzle translating from English to those languages in the same family as {name}. The puzzles that you generate should be distinct from one another than the example puzzles, and the test puzzle but should help establish the relationships for translation between {name} and English. They also should be from a diverse set of languages within the same language family as the test puzzle. Provide your outputs in the format of **[your answer]**."

#### F.7.3 DEDUCTION STEP PROMPT

"This is a translation puzzle. In a moment, you will use logic and analytical reasoning to translate from a never-seen-before language ({name}) to English. Your task is to look closely at the example puzzles and the puzzles that you have generated in order to solve the test puzzle. Take a deep breath and work on this problem step-by-step in a logical way, using careful analytical

reasoning to get the correct result. When you are done with your answer, provide your outputs in the format of **[your answer]**.”

## G    ORACLE LANGUAGE FAMILIES

Table 12: Oracle language families used for the results in Table 4, where we present a language family label to the model rather than (implicitly) instructing it to infer the language family.

| Target Language | Oracle Language Family |
| --- | --- |
| Abun | West Papuan |
| Ainu | Ainu / Language Isolate |
| Ayutla Mixe | Mixe-Zoque |
| Bangime | Language Isolate |
| Chimalapa Zoque | Mixe-Zoque |
| Dogon | Niger-Congo |
| Engenni | Niger-Congo |
| Guugu Yimithirr | Pama-Nyungan |
| Kalam | Kalam |
| Komi-Ziran | Uralic |
| Kutenai | Language Isolate |
| Mapudungan | Araucanian |
| Misantla Totonac | Totonacan |
| Mixtepec Zapotec | Oto-Manguean |
| Ngadha | Austronesian Malayo-Polynesian |
| Niuean | Malayo-Polynesian |
| Rapa Nui | Austronesian Malayo-Polynesian |
| Seri | Hokan / Language Isolate |
| Totonac | Totonacan |

## H    LANGUAGE IDENTIFICATION IN ANALOGICAL PROMPTING WITH INFERRED FAMILIES

We analyze the ability for frontier models (GPT-4o, Llama-3.1-405B-Instruct) to produce the correct language family labels solely by being prompted to produce exemplars in the same language family. The results for GPT-4o are included in Table 10, and the results with Llama-3.1-405B-Instruct are included in Table 11. The phrase ”synthetic” is used as a catch-all for the model determining that the language is ”constructed”, ”synthetic”, ”fictional”, ”hypothetical”, or any similar synonym. There are some instances where the model does not produce any label for the language family, and begins immediately producing exemplar puzzles from some implicitly chosen set of languages, without stating that list; this is listed in the tables as ”None”. For Language Isolates that are debated (e.g. Seri, which is considered an isolate by some linguists, and a member of the Hokan language family by others), we specify which label was provide, but assign either as correct when determining each model’s correctness rate. Furthermore, the model may not necessarily produce the leaf-level language family, but rather, a larger family which includes the leaf-level one (e.g. Trans-New Guinea instead of Kalam, which belongs to the Trans-New Guinea family).

Table 13: Inferred language families by Llama-3.1-405B-Instruct, where the model is prompted in our 2-stage approach to first produce exemplars in the same language family and then apply them to solve the test phrase. The model often identifies the language family which the target language is a member of ("label") which we report below, prior to identifying languages within that family, that are geographically proximal, or if the model predicted that it is an isolate or believes the language to be synthetic, produces similar *synthetic* languages.

| Target Language | Number of Questions | Inferred Language Family |
|---|---|---|
| Abun | 4 | West Papuan (4) |
| Ainu | 8 | Language Isolate (8) |
| Ayutla Mixe | 4 | Mixe-Zoque (4) |
| Bangime | 36 | Isolate (25), Niger-Congo (11) |
| Chimalapa Zoque | 12 | Zoquean (12) |
| Dogon | 8 | Niger-Congo (6), None (2) |
| Engenni | 25 | Niger-Congo (25) |
| Guugu Yimithir | 10 | Pama-Nyungan (10) |
| Kalam | 6 | Trans-New Guinea (6) |
| Komi-Ziran | 6 | Uralic (6) |
| Kutenai | 5 | Language Isolate (5) |
| Mapudungan | 24 | Araucanian (14), Synthetic (10) |
| Misantla Totonac | 4 | Totonacan (4) |
| Mixtepec Zapotec | 24 | Oto-Manguean (24) |
| Ngadha | 14 | Austronesian (14) |
| Niuean | 18 | Polynesian (18) |
| Rapa Nui | 37 | Polynesian (37) |
| Seri | 21 | Hokan / Isolate (17), Isolate (4) |
| Totonac | 6 | Totonacan (6) |

Our analysis reveals that both models are quite adept at identifying language families reliably. In fact, Llama-3.1-405B-Instruct's language family correctness out of the 272 samples, relative to the oracle labels in Appendix F is an astounding $\frac{249}{272} = 91.54\%$, while GPT-4o's rate is $\frac{202}{272} = 74.26\%$.

We report anecdotally that while both models appear to have a strong understanding of the leaf-level language families (e.g. the Edoid family), Llama-3.1-405B-Instruct seems to have a stronger *taxonomical* understanding, producing outputs such as "Chimalapa Zoque is a member of the Zoquean branch of the Zoque-Tzeltalan language family, which is part of the larger Mayan language family." By contrast, GPT-4o often would solely identify the direct parent of the language in question, producing outputs such as "Chimalapa Zoque belongs to the Mixe-Zoque language family." It appears that by the statements made at the start of the response, GPT-4o appears to (at least claim to) base its choice of language family based on the structure of the source-target provided exemplar translations, such as the following: "Based on the examples provided in Mapudungan 3, it seems to encode simple noun phrases with an adjective-noun structure. To generate similar puzzles from other languages potentially in the same family (Araucanian), we should maintain this structure and ensure variety in the adjectives and nouns used." Similarly, it produces statements such as "Based on the examples provided in Rapa Nui, I can infer common Polynesian morphological and syntactical patterns that will help in generating puzzles for other related languages within the Austronesian language family, specifically the Polynesian subfamily."

Furthermore, through the process of obtaining the counts in the tables listed here, we observed that both models struggled when it was specified that there were multiple separate problems for a given language. For instance, both models do not struggle much with identifying the correct language family for "Mapudungan 1" as Araucanian, but completely either fail to identify any language family (GPT-4o) or suggest that the language is synthetic when given "Mapudungan 4". This is an interesting phenomenon that we propose merits further study.

Table 14: Inferred language families by GPT-4o.

| Target Language | Number of Questions | Inferred Language Family |
|---|---|---|
| Abun | 4 | West Papuan (3), Lakes Plain (1) |
| Ainu | 8 | Language Isolate (8) |
| Ayutla Mixe | 4 | Mixe-Zoque (4) |
| Bangime | 36 | Isolate (18), Niger-Congo (2), Synthetic (16) |
| Chimalapa Zoque | 12 | Mixe-Zoque (12) |
| Dogon | 8 | Niger-Congo (6), Isolate (1), None (1) |
| Engenni | 25 | Niger-Congo (21), Synthetic (2), None (2) |
| Guugu Yimithir | 10 | Pama-Nyungan (10) |
| Kalam | 6 | Trans-New Guinea (5), Austronesian (1) |
| Komi-Ziran | 6 | Uralic (4), Synthetic (2) |
| Kutenai | 5 | Language Isolate (5) |
| Mapudungan | 24 | Araucanian (3), Synthetic (3), None (18) |
| Misantla Totonac | 4 | Totonacan (4) |
| Mixtepec Zapotec | 24 | Oto-Manguean (24) |
| Ngadha | 14 | Austronesian (14) |
| Niuean | 18 | Polynesian (16), Synthetic (1), None (1) |
| Rapa Nui | 37 | Polynesian (30), Synthetic (3), None (4) |
| Seri | 21 | Isolate (6), Hokan (3), Synthetic (6), None (6) |
| Totonac | 6 | Totonacan (6) |

# I    ABLATIONS WITH LLAMA-3.1-8B-INSTRUCT

We also examine the performance of another weak model, namely Llama-3.1-8B-Instruct. This model achieves similar performance on the baseline experiments as Aya-35B, and despite not being a specialized multilingual model like Aya, has seen 15T tokens of multilingual pre-training data, as well as large volumes of multilingual SFT and post-training data, leveraging human annotations by a constructed multilingual expert pre-trained model. We report these results in a 3x3 grid as in Section 4.2, where the model on the left side is the analogical exemplar generator, and the right hand side is the model which applies inductively learned rules; this includes the self-generation (diagonal), inference-time distillation, and weak-to-strong settings. Note that the results of the top left 2x2 (between GPT-4o and Llama-3.1-405B-Instruct) are the same as those reported in Section 4.2.

Table 15: The results of Table 2, mixing-and-matching the generator and deducer models, with Llama-3.1-8B-Instruct in place of Aya-35B.

| Deducer
Generator | GPT-4o | Llama-3.1-405B-Instruct | Llama-3.1-8B-Instruct |
|---|---|---|---|
| GPT-4o | 66.91% | 71.69% | 22.30% |
| Llama-3.1-405B-Instruct | 67.28% | 67.65% | 19.12% |
| Llama-3.1-8B-Instruct | 63.36% | 70.96% | 20.10% |

Like Aya-35B, Llama-3.1-8B-Instruct does not improve with inference-time exemplar distillation. However, despite smaller gains (4.2% over baseline) in the weak-to-strong setting with GPT-4o as the deducer, we achieve nearly 71% with Llama-3.1-405B as the deducer. This further reinforces the notion that Llama-3.1-405B is the strongest current model at inductive and deductive reasoning, as it attains higher results than the next best model, GPT-4o, across all analogical generator models.

## J 1-STAGE ANALOGICAL PROMPTING

We study the 1-stage analogical prompting setting as posed in Yasunaga et al. (2024), where analogical exemplars are generated and applied through the same instruction, all at once.

Table 16: Results with 1-stage analogical prompting (where both generation and application occur through a single instruction).

| Model | 1-Stage Analogical Prompting |
|---|---|
| GPT-3.5-Turbo | 2.21% |
| GPT-4 | 34.93% |
| GPT-4o | 38.60% |
| Llama-3.1-8B-Instruct | 3.31% |
| Llama-3.1-70B-Instruct | 27.21% |
| Llama-3.1-405B-Instruct | 22.43% |
| Mixtral-8x7B-Instruct | 1.1% |
| Mixtral-8x22B-Instruct | 34.56% |

From our error analysis, we observe that even our strongest models such as GPT-4o are confused by the 1-stage analogical reasoning prompt. That is, prompting models to identify the language family of the test sample, identify multiple languages in that family, produce several puzzles of exemplars translating to and from English to those languages such that they are sufficiently diverse from one another, and apply all of the exemplars to the test puzzle made for an overloaded instruction. Splitting the instruction into 2 stages – generating analogical exemplars, then prompting with both the provided and generated exemplars – is a natural solution. Evidently, as shown in Table 2, using 2-stage analogical prompting proves effective.

