# OpenReview forum: "Inductive Linguistic Reasoning with Large Language Models"
_ICLR.cc/2025/Conference — Submitted to ICLR 2025_

### Official Review · Reviewer_KXR4 · 2024-10-25

**Soundness:** 2
**Presentation:** 3
**Contribution:** 2
**Rating:** 1
**Confidence:** 4

**Summary:**

The paper explores the capabilities of LLM in performing linguistic reasoning on low-resource languages through language puzzles. This study uses the ‘analogical prompting’ approach, which enhances the reasoning capabilities of these models by using analogy-generated examples to improve performance in translation tasks, particularly in low-resource languages.

The idea is very interesting, and this is the first contribution that transfers the idea beyond English. However, there are some really serious points that emerge (detailed below). This does not put the paper in a good light and it strongly needs revision.

**Strengths:**

The idea is interesting because using the analogical reasoning approach proposed in ‘Large language models as analogical reasoners’ on multilingual tasks is a methodology that has been little explored and apparently shows promise.


The authors propose a good experimental setting and a comprehensive discussion, however with difficulty one understands some passages

**Weaknesses:**

Among the paper's weaknesses are:

- The image Figure 1 is very confusing and is really difficult to read as it is of very poor quality.

- Many steps should be carefully explained e.g. the heart section (section 2 introduces the method that should emulate multilingual analogical reasoning). No examples are given in this section and the problem is not formalised, confusing the reader.

- The experiments, although many, are poorly introduced and the thread is not understood.

**Questions:**

How did you conduct the evaluation?

Do you plan to release the code publicly?

---

> ### Author Response · Authors · 2024-11-23
> **Response to Reviewer KXR4**
>
> We would like to thank Reviewer KXR4 for taking the time to review our paper. We appreciate that the reviewer agrees with the value and novelty of our contributions, and for finds our work to have a “good experimental setting and a comprehensive discussion”. We address the points raised in the review in our responses below, and in our revised paper.
>
> > The image Figure 1 is very confusing and is really difficult to read as it is of very poor quality.
>
> We’ve incorporated this feedback to revise Figure 1 to be clearer and more directly illustrative of our method, hopefully alleviating any confusion.
>
>
> > Many steps should be carefully explained e.g. the heart section (section 2 introduces the method that should emulate multilingual analogical reasoning). No examples are given in this section and the problem is not formalised, confusing the reader.
>
> Thank you for this feedback -- we’ve now included a discussion of the translation setting of interest in linguistics olympiad problems (“Rosetta Stone” problems) in Section 2, along with an example of such a problem. Figure 1 also illustrates our 2-stage analogical reasoning method, and we address this in the text of Section 2 as well.
>
>
> > The experiments, although many, are poorly introduced and the thread is not understood.
>
> We appreciate this feedback, and have sought to incorporate it to improve our paper. We have added pointers in Section 3 to the respective results sections (in Section 4) where they are addressed. We hope that this makes the thread of experiments easier to follow. Here is the outline of our core experiments:
> 1. Section 4.1 consists of 4 baseline experiments (zero-shot, few-shot, few-shot with CoT prompting, and few-shot with CoT prompting to induce a full rationale) introduced in Section 3.1.
> 2. Section 4.2 first compares the 3 methods introduced in Section 3.2. We then compare the best self-generated results of that experiment ("inferred families") against providing a language family label ("oracle families").
>
> > How did you conduct the evaluation?
>
> Evaluation is performed for exact match as the primary metric, due to the noted fallacies of ChrF2 and corpus-level BLEU scores in Section 4. All 272 problems were manually evaluated by one of the authors of this work; the annotation was purely for exact match (without partial scoring or other subjective notions for which inter-annotator agreement would prove as a useful signal). The sole necessity of human evaluation in using exact match is due to parsing errors in instruction following that we find with smaller models like Llama-3.1-8B and Aya-8B; to keep the evaluation protocol consistent, this was repeated for all experiments.
>
> > Do you plan to release the code publicly?
>
> Yes, we will release the code publicly in the camera-ready release, for the community at large to apply our method as well as to guide future efforts in linguistic reasoning with LLMs in IOL-style problems. Furthermore, given our method is an inference-time intervention to boost performance through prompting, the prompts included in Appendix D should suffice for the purpose of reproducibility.
>
> ---
>
> We hope that our revised draft and these responses address your concerns. We would be glad to address any further concerns.

---

> > ### Author Response · Authors · 2024-11-26
> > **Official Comment by Authors (Friendly Reminder)**
> >
> > Dear Reviewer KXR4,
> >
> > Thank you very much again for your helpful feedback. We have carefully responded to your concerns and questions, and incorporated them into our revised paper. As the revision period is ending soon, we would greatly appreciate your feedback on our responses and revision. If our response has resolved your concerns, we would like to respectfully ask you to consider raising the score for our work. Thank you again for your time!

---

> > > ### Author Response · Authors · 2024-12-02
> > > **Friendly Reminder: Discussion Period Deadline**
> > >
> > > Dear Reviewer KXR4,
> > >
> > > Thank you again for your valuable feedback on our initial version. As the discussion period will be ending shortly, we would be glad to address any remaining questions; if our revisions and responses have resolved your concerns, we would greatly appreciate it if you could consider improving the evaluation of our work. Thank you very much again for your time and consideration!

---

### Official Review · Reviewer_SDWx · 2024-10-31

**Soundness:** 2
**Presentation:** 3
**Contribution:** 2
**Rating:** 5
**Confidence:** 4

**Summary:**

The paper explores analogical prompting to solve modeLing (Chi et al., 2024), a dataset containing International Linguistics Olympiad-style problems. Through experiments with proprietary and open-source models using different prompting strategies, the authors demonstrate that few-shot chain-of-thought prompting with explanatory rationales yields optimal performance. They further suggest including analogical exemplars ( language family information obtained through LLM prompting) in prompts can enhance model performance.

**Strengths:**

1. There are limited works on solving Linguistics Olympiad problems.  This paper's methodology is valuable as a benchmark for future studies.
2. The study presents comprehensive experiments across various models and prompting techniques, with a clear presentation of results.

**Weaknesses:**

1. The paper's contribution is primarily empirical, with limited conceptual innovation. The approach of using analogical prompting to boost performance is not very inspiring, as it mainly involves augmenting prompts with self-generated information [1].

2. The authors tested their method only on machine translation tasks, overlooking other question formats in IOL, such as multiple-choice and cloze questions. A more suitable benchmark than modeLing would be [2] or [3].

3. It is widely known that closely related languages help with cross-lingual transfer [4] [5]. This paper, however, does not seem to provide any novel insights in this area.


References:

[1] Sun, Z., Wang, X., Tay, Y., Yang, Y., & Zhou, D. (2022). Recitation-Augmented Language Models. ICLR 2023.

[2] Sánchez, E., Alastruey, B., Ropers, C., Stenetorp, P., Artetxe, M., & Costa-jussà, M. R. (2024). Linguini: A benchmark for language-agnostic linguistic reasoning. arXiv preprint arXiv:2409.12126.

[3] Bean, A. M., Hellsten, S., Mayne, H., Magomere, J., Chi, E. A., Chi, R., ... & Kirk, H. R. (2024). LINGOLY: A Benchmark of Olympiad-Level Linguistic Reasoning Puzzles in Low-Resource and Extinct Languages. arXiv preprint arXiv:2406.06196.

[4] Dan Malkin, Tomasz Limisiewicz, and Gabriel Stanovsky. 2022. A Balanced Data Approach for Evaluating Cross-Lingual Transfer: Mapping the Linguistic Blood Bank. NAACL 2022.

[5] Vésteinn Snæbjarnarson, Annika Simonsen, Goran Glavaš, and Ivan Vulić. 2023. Transfer to a Low-Resource Language via Close Relatives: The Case Study on Faroese.NoDaLiDa 2023.

**Questions:**

1. In Table 1, zero-shot scores are near zero across all models, which is unexpected, given that BLEU metrics are relatively lenient. Any insights into why this might be the case?

2. Line 311-313: "Our findings suggest that when equipped with the right tools (analogical demonstrations) from effective multilingual reasoners, strong deducers can thrive.". However, in Table 2, using Aya-23-35B as the generator yields better results than Llama-405B (which performed better in prior evaluations) when GPT-4o is the deducer. Does this imply that Aya excels at language identification rather than machine translation?

---

> ### Author Response · Authors · 2024-11-23
> **Response to Reviewer SDWx (Part 1/2)**
>
> We would like to thank Reviewer SDWx for taking the time to review our paper and for their valuable feedback. We appreciate that the reviewer agrees that our work’s methodology is “valuable” and has “comprehensive experiments” with “clear presentation of results”. Our responses below address the concerns and questions raised in the review.
>
> > The paper's contribution is primarily empirical, with limited conceptual innovation. The approach of using analogical prompting to boost performance is not very inspiring, as it mainly involves augmenting prompts with self-generated information [1].
>
> While our method is inspired by analogical prompting, we note that it differs from the evaluation in Yasunaga et al. 2023, by the “distance” from the target problem to those likely seen in these model’s training corpora. For instance, in Yasunaga et al. 2023 [1], which evaluates on mathematical reasoning tasks in the GSM8K and MATH datasets, these models have seen (similar) math problems in both the pre-training and supervised fine-tuning corpora. In Sun et al. 2022 [2], the tasks are knowledge-intensive closed-book question-answering questions in English (NQ, TriviaQA, and HotpotQA), where the answers to similar problems or relevant facts could be expected to be in the model’s knowledge base. In the context of extremely low-resource languages, however, as addressed in our work, prompting for another language in the same family does not consistently yield a high-resource language, per se, or an instance we would necessarily expect the model to have in its parametric knowledge. We have included an analysis on the language families identified (or rather, the similar languages that the model generates on) in Appendix G of our revised paper.
>
> > The authors tested their method only on machine translation tasks, overlooking other question formats in IOL, such as multiple-choice and cloze questions. A more suitable benchmark than modeLing would be [2] or [3].
>
> We focus on machine translation tasks -- more specifically, “Rosetta Stone” puzzles -- as these are the primary focus of both the PuzzLing Machines and modeLing datasets as noted in Sections 3.3 and 5.2. Furthermore, the “Rosetta” category constitutes the largest percentage of the problems in LINGOLY, at $46\\%$ of the problems. The problems in LINGOLY are drawn from the UK Linguistics Olympiad (UKLO), and as such may still be susceptible to leakage, in contrast to modeLing, which presents an entirely unseen set of questions. This is reflected in examining our zero-shot results (which never exceed $1.5\\%$ across all models) and the “no context” baseline in LINGOLY, which can be inferred from their exact match and $\\Delta_{NC}$ scores, which is much higher. This would either be attributable to leakage or the models being familiar with (at least some of) the languages being tested upon.
>
> Thank you for bringing the Linguini benchmark to our attention — this was concurrent work that we were unaware of, which appears to have been released shortly (one week) before the ICLR deadline.  As we note in section 3, we do not intend to claim that our work addresses all problem types from the IOL competition, which may also require multimodal inputs, but our scope lies with these “Rosetta Stone” puzzles. To this effect, we have added Section 2.1 to discuss the nature of the problems of interest and provide an example of such a problem. Nonetheless, keeping in spirit with expanding the generalizability of our findings, we are currently working on evaluating our method on the LingOly dataset, per your suggestion, which we will include in the camera-ready version.
>
>
> > It is widely known that closely related languages help with cross-lingual transfer [4] [5]. This paper, however, does not seem to provide any novel insights in this area.
>
> While we acknowledge that there are other works that study the impact of closely-related languages for zero-shot transfer, our work does not specify any similar languages in the instruction nor does it perform any training conditional on this knowledge. A key insight of our work is that models can automatically identify exemplars in similar, seen languages, and in fact can do so even better than when a language family label is specified in the prompt (oracle families). This finding is novel (to the best of our knowledge), and as such, differs from prior art. Furthermore, our findings in language models being able to identify the language families of extremely low-resource or nearly extinct languages at a high rate at test-time without any additional training to do so (included in Appendix G of our revised draft), and even constructing synthetic languages similar to language isolates, is a new contribution as far as we are aware.

---

> > ### Comment · Reviewer_SDWx · 2024-11-27
> > **response to authors**
> >
> > Thank you for your detailed responses and revisions. I appreciate the expanded discussion in Section 2.1 and Appendix G, and your plans to evaluate on LingOly. However, I still have some concerns:
> >
> > 1. The automatic identification of related languages is interesting, but the approach remains largely empirical. In addition, focusing solely on machine translation limits the applicability of your approach to other IOL tasks. A preliminary evaluation of benchmarks like LingOly or Linguini in a revised version would strengthen the paper.
> >
> > 2. The claim of novelty in cross-lingual transfer lacks sufficient evidence. More detailed quantitative analysis or expert validation of the generated exemplars would help substantiate this.
> >
> > Overall, your work provides useful empirical insights, and I have raised my score to 5, but further emphasis on novelty and generalizability would enhance its impact.

---

> ### Author Response · Authors · 2024-11-23
> **Response to Reviewer SDWx (Part 2/2)**
>
> > In Table 1, zero-shot scores are near zero across all models, which is unexpected, given that BLEU metrics are relatively lenient. Any insights into why this might be the case?
>
> Please note that the scores in Table 1 are exact match, not corpus-level BLEU; these are instead included in Appendix A.2. Nonetheless, it is true that the BLEU scores are still near-zero — we find that all models rarely get even a single token correct in the zero-shot setting due to the challenge of these problems. We find this to be another proof of the fact that translation examples of these extremely low-resource languages have not been seen by the model in either pre-training or post-training.
>
> > Line 311-313: "Our findings suggest that when equipped with the right tools (analogical demonstrations) from effective multilingual reasoners, strong deducers can thrive.". However, in Table 2, using Aya-23-35B as the generator yields better results than Llama-405B (which performed better in prior evaluations) when GPT-4o is the deducer. Does this imply that Aya excels at language identification rather than machine translation?
>
> This is a great question, and it is challenging to compare the language identification and (general) translation skills of the model. In Appendix G, we introduce a table examining the language families which models identified in producing analogical exemplars. However, we only include GPT-4o and Llama-3.1-405B-Instruct, due to Aya-35B never producing an explicit language family label. That is, while Aya-35B produces analogical exemplars from the same language family as the target language (e.g. "Here are some puzzles translating from and to languages in the same family as Chimalapa Zoque:”), it doesn’t produce the Mixe-Zoque language family as a label anywhere in its response, unlike the other models. However, we do conclude that for the purpose of solving these puzzles, Aya-35B seems to hold more utility for language identification in producing such exemplars (although, as we note, it is challenging to assess the quality of these exemplars without expert annotators), due to the success of the weak-to-strong method, than it does as a deducer in translation.
>
> ### References
>
> [1] Yasunaga, M., et al. (2023). Large Language Models as Analogical Reasoners. ICLR 2024.
>
> [2] Sun, Z., et al. (2022). Recitation-Augmented Language Models. ICLR 2023.
>
> ---
> We hope that our response and revised paper addresses your concerns and questions. We would be happy to address any further concerns.

---

> > ### Author Response · Authors · 2024-11-26
> > **Official Comment by Authors (Friendly Reminder)**
> >
> > Dear Reviewer SDWx,
> >
> > Thank you very much again for your valuable feedback. We have carefully responded to your concerns and questions, and incorporated them into our revised paper. With the revision deadline approaching soon, we would greatly appreciate your feedback on our responses and revision, which we hope have addressed your concerns and clarified the points raised. If so, we would like to respectfully ask you to reconsider your assessment; we would also be happy to address any further concerns you may have. Thank you again for your time!

---

> ### Author Response · Authors · 2024-11-28
> **Response to Reviewer SDWx (Part 1/2)**
>
> Thank you for your reply, and for the feedback! We include preliminary results on the LINGOLY dataset below; these apply our 2-stage analogical prompting approach with GPT-4o. We compare against the baselines with the respective models as reported in Bean et al. 2024 [1]. The table below is reflected in figure format in Appendix D in our revised paper. We also provide difficulty / category element-wise differences over the baseline, which highlight the improvements yielded through our method.  Note that breakthrough is the easiest set of problems and Round 2 is the most difficult, which serves as the UK invitational qualification exam for the IOL. The results below correspond to the exact match scores reported in LINGOLY. Please note that all empty cells correspond to combinations for which problems do not currently exist in the LINGOLY dataset.
>
> Baseline Results for GPT-4o (reported in LINGOLY):
>
> |              | Computational | Text | Monolingual | Match-up | Pattern | Rosetta |
> | :------------: | :-------------: | :----: | :-----------: | :--------: | :-------: | :-------: |
> | Breakthrough |               | 100% |             |          | 47%     | 79%     |
> | Foundation  |     0%     |      |             | 100%     | 67%     | 62%     |
> | Intermediate |               |      |             |          | 58%     | 34%     |
> | Advanced     |               |      | 0%          | 33%      | 53%     | 26%     |
> | Round 2      |               |      | 0%          | 30%      | 27%     | 12%     |
>
>
> Our Results with GPT-4o, 2-stage analogical prompting:
>
> |              | Computational | Text | Monolingual | Match-up | Pattern | Rosetta |
> | :------------: | :-------------: | :----: | :-----------: | :--------: | :-------: | :-------: |
> | Breakthrough |               | 100% |             |          | 80%     | 86%     |
> | Foundation   | 0%            |      |             | 100%     | 69%     | 80%     |
> | Intermediate |               |      |             |          | 83%     | 64%     |
> | Advanced     |               |      | 19%         | 50%      | 73%     | 51%     |
> | Round 2      |               |      | 14%         | 42%      | 49%     | 41%     |
>
>
> Deltas (Difference between our result and baseline):
>
> |              | Computational | Text | Monolingual | Match-up | Pattern | Rosetta |
> | :------------: | :-------------: | :----: | :-----------: | :--------: | :-------: | :-------: |
> | Breakthrough |               | 0%   |             |          | +33%    | +7%     |
> | Foundation   | 0%            |      |             | 0%       | +2%     | +18%    |
> | Intermediate |               |      |             |          | +25%    | +30%    |
> | Advanced     |               |      | +19%        | +17%     | +20%    | +25%    |
> | Round 2      |               |      | +14%        | +12%     | +22%    | +29%    |
>
>
> Encouragingly, we find that our results significantly outperform the baseline by a sizable amount across all difficulty levels, and across all tasks. Moreover, the results outperform the Claude-3 Opus state-of-the-art scores reported in the LINGOLY paper on every single setting, with the exception of the Breakthrough Rosetta Stone (which are the easiest problems). Specifically, we find that our 2-stage analogical prompting method enables GPT-4o to solve questions of the monolingual type which it could not before (0% —> 19% and 14%); furthermore, the correctness rates jump considerably for some of the hardest categories over the baseline (1.81x improvement in Round 2 Pattern, 1.96x in Advanced Rosetta Stone, and 3.42x in Round 2 Rosetta Stone). It is especially worth noting that the Round 2 Rosetta Stone results corroborate with our findings on modeLing as reported in our paper. **These findings suggest that our method generalizes across both datasets and question types.**
>
> A note on the prompting method: we use the same prompts as before, with the addition of the preamble and context provided in LINGOLY for all of the tasks. The context includes both the background on the problem and the exemplars in the target language, as in our evaluation with modeLing. As such, there might be opportunities to further improve the results by optimizing the prompts for the various other tasks introduced in this dataset.
>
> We have added a note in Section 3.3 of the question types included in LINGOLY, which expand beyond the "Rosetta Stone" translation problem style (although, as aforementioned, the Rosetta Stone category still constitutes a sizable fraction of the problems). Furthermore, these preliminary results have been added to a new appendix section (Appendix D; shifting the other sections accordingly) in a revision of our paper which we have uploaded.

---

> ### Author Response · Authors · 2024-11-28
> **Response to Reviewer SDWx (Part 2/2)**
>
> **Expert Validation**
>
> On the point of expert validation — we refer the reader to our limitations, Section 2 on exemplar correctness, and in footnote 3, where we discuss the challenge in introducing a single reliable expert who would be sufficiently familiar with and adept at understanding the typological rarities present in these extremely low-resource languages. Most of the languages in modeLing do not have a consolidated grammar book, to the best of our knowledge (unlike MTOB [2], for which there exists a grammar book for Kalamang), from which one could verify correctness. While we appreciate and acknowledge the importance of a forming a deeper, concept-level understanding of what occurs in the cross-lingual transfer from auxiliary language exemplars, we posit that this would likely require either a global crowd-sourcing initiative (which is practically challenging) or a interpretability-driven analysis to study concept learning, which we find to be beyond the scope of our work.
>
> **References**
>
> [1] Bean, A. M., Hellsten, S., Mayne, H., Magomere, J., Chi, E. A., Chi, R., Hale, S. A., & Kirk, H. R. (2024). LINGOLY: A Benchmark of Olympiad-Level Linguistic Reasoning Puzzles in Low-Resource and Extinct Languages. arXiv preprint arXiv:2406.06196.
>
> [2] Tanzer, G., Suzgun, M., Visser, E., Jurafsky, D., & Melas-Kyriazi, L. (2024). A Benchmark for Learning to Translate a New Language from One Grammar Book. arXiv preprint arXiv:2309.16575v2.
>
> ----
>
> Thank you very much again for your feedback, and we hope our promising results above and responses have addressed your concerns!

---

> > ### Author Response · Authors · 2024-12-02
> > **Friendly Reminder: Discussion Period Deadline**
> >
> > Dear Reviewer SDWx,
> >
> > Thank you again for your valuable feedback, and we hope that the results shared above and added to our revised paper address the concerns raised. As the discussion period will be ending shortly, we would be glad to address any remaining questions or concerns. We would greatly appreciate it if you could reconsider the evaluation of our work. Thank you very much again for your time and consideration!

---

### Official Review · Reviewer_uieZ · 2024-10-31

**Soundness:** 2
**Presentation:** 2
**Contribution:** 2
**Rating:** 5
**Confidence:** 4

**Summary:**

This paper investigates the possibility of using few shot learning so that LLMs can generalize their knowledge to new and highly under resourced languages at inference time.  The authors introduce a new method of prompting called 2 stage analogical prompting, according to which for a given language problem P for a novel or very under resourced language L they first get a model to infer what family of languages L belongs to, then they get the model to select languages in L's family and to produce language problems similar to the given one P.  These results are then fed into either the same or a different model to then solve the original language problem P.  The authors show that their 2 stage analogical prompting delivers superior results to other  state of the art prompting (CoT) and methods without CoT.

**Strengths:**

The 2 stage analogical prompt is interesting and suggests that perhaps models might leverage information about related but more represented languages to solve the given linguistic problems in the test set.  There is also an interesting difference between larger models like Llama 405B or GPT4o and smaller models; the analogical exemplars work for the larger models but not the smaller ones, pointing to an ability of the larger models to adapt the analogical examples to the given linguistic problem that the smaller models lack.

**Weaknesses:**

The paper's main weakness is the disconnect between the empirical investigations, which seem sound enough, and the desired conclusion that is given here: "In summary, our results suggest that the ability of the model to deduce from inductively learned rules is the key performance driver."  In other parts of the paper the rules referred to here would seem to be grammar rules.  There is little in the paper to suggest in the results that any grammar rules have been really learned or what the form of the grammar rules might be.  For example the rules could involve simple agreement or complex long distance effects governing ellipsis, gapping, or some other complex grammatical phenomenon.   At least this reviewer would like to see a much more detailed study in which (i) the grammar rules at issue are clearly stated, (ii) we have results for patterns that are governed by the rules (iii) we have results for constructed examples that violate those rules.  I would expect that for examples that violate the rules the models would either fail to produce an output or flag it in some way, if they had learned the grammatical rules.  The paper provides no such data, and so we can't really conclude anything about the mechanism that the models used to infer correct solutions to the language problems posed.

Another problem with this paper is the reference to linguistic problems that aren't really very well described.  One can gather that at least some of the problems are translation problems.  But are they all translation problems?  If so, how on an olympiad test would a participant be able to get a good translation for a completely unknown language without any clues?   The whole experimental basis of the paper is kind of murky and needs to be cleaned up for those readers who are unfamiliar with the linguistic olympiads.

The strengths of the paper could be improved by looking into more detail as to what 2 step analogical reasoning is doing.
I might of missed it but it seems that the paper itself doesn't contain a discussion of what happens when the language family is omitted but the examples are provided.  It would have been nice to have a more detailed stucy of the analogical reasoning itself.

**Questions:**

Please describe in more detail the test linguistic problems in this study.

What are rationales in the particular case of linguistic problems?

---

> ### Author Response · Authors · 2024-11-23
> **Response to Reviewer uieZ (Part 1/2)**
>
> We would like to thank Reviewer uieZ for taking the time to review our paper, for their valuable feedback, and for finding our 2-stage analogical prompting method “interesting”. We address the concerns raised in the review below and in our revised paper.
>
> > There is little in the paper to suggest in the results that any grammar rules have been really learned or what the form of the grammar rules might be.
>
> This is very helpful feedback, and we appreciate the reviewer’s thoughtful response on this point. We acknowledge that the usage of the term “rules” as used in our statement does not quite imply that grammar rules have been rigorously learned, as the reviewer has pointed out. What we intend to say as a conclusion, rather, is on the basis of the evidence that these frontier models first produce token-level mappings between the source and target language for the few-shot exemplars, and token-level mappings between the source and auxiliary languages for the analogical exemplars; then apply these mappings to the test phrase. That is, these mappings are what we intend to refer to as “rules”. Nonetheless, this point is well-taken, and we have rephrased this conclusion accordingly to avoid confusion.
>
>
> > One can gather that at least some of the problems are translation problems. But are they all translation problems? If so, how on an olympiad test would a participant be able to get a good translation for a completely unknown language without any clues? The whole experimental basis of the paper is kind of murky and needs to be cleaned up for those readers who are unfamiliar with the linguistic olympiads.
>
> We focus solely on machine translation tasks -- more specifically, “Rosetta Stone” puzzles -- as these are the primary focus of the modeLing dataset as noted in Sections 3.3 and 5.2. Indeed, in these linguistics olympiad tests, participants are expected to perform translations for often-unseen languages, solely given a list of examples from which to infer surface-level associations and apply pattern matching (deductive reasoning) to solve. We appreciate the feedback on the nature of these puzzles being unclear, and have updated our paper to include Section 2.1, a discussion on the problems of interest in our work and an example of such a puzzle.
>
> > I might of missed it but it seems that the paper itself doesn't contain a discussion of what happens when the language family is omitted but the examples are provided. It would have been nice to have a more detailed stucy of the analogical reasoning itself.
>
> (Repeated from general response): We analyze the language family labels produced by Llama-3.1-405B-Instruct and GPT-4o in the first stage of our analogical prompting procedure, prior to identifying specific languages from said family and producing the analogical exemplars. This corresponds to the inferred language families experiments in Figure 2b, where the model is only prompted to produce puzzles from other languages in the same family, *without specifying that family*. We compare these labels against the oracle labels in the table included in Appendix F, to yield a correctness score; we include these tables in Appendix G.
>
> We also qualitatively discuss models’ behavior in 2-stage analogical reasoning, added to Section 4.2, under “Analogical reasoning boosts frontier models” in our revised draft (repeated here): In the first stage, both of these frontier models correctly identify the language family at a fairly high rate (see tables above), select a few languages from said family, and generate analogical puzzles for those auxiliary languages, as intended. Then, in the second stage, the model walks through the tokens in the test phrase, and analyzes how each is to be translated to the target languages, and then combines them together in the appropriate order based on following the syntactical patterns observed from the given exemplars. Thus, it appears that the model uses the analogical exemplars to better induce the mappings of words in the target language to words in the source language, which it then applies to the target phrase.

---

> ### Author Response · Authors · 2024-11-23
> **Response to Reviewer uieZ (Part 2/2)**
>
> > What are rationales in the particular case of linguistic problems?
>
> Rationales in the context of these translation task involve word-by-word explanations behind why a word in English maps to one in the target language (or vice versa), as well as any explanations regarding the ordering of words. Here is the exemplar provided for the few-shot chain-of-thought with rationale prompt setting (also included in Appendix E.5):
>
>     1. Spanish: ventana roja English: red window
>     2. Spanish: ventana azul English: blue window
>     3. Spanish: manzana azul English: blue apple
>     Using the above examples, translate the following.
>     Spanish: manzana roja
>     EXPLANATION: The first step we notice is that the word “ventana” must mean window because (1) the word “ventana” appears twice between sentences 1 and 2, and (2) the only word that appears twice in the English translation is “window.” Next, we infer that “roja” must be “red” and “azul” must be “blue” by process of elimination. Next, we guess that in Spanish, the noun precedes the adjective because “ventana” comes before “roja” and “azul.” Therefore, the noun in sentence 3 (“apple”) must correspond to the word preceding the adjective (“manzana”) in the Spanish translations. Putting this together, “manzana roja” must mean “red apple” in English.
>
>     ANSWER: English: red apple.
>     Now, given the following test phrase, please translate it. Take a deep breath and work on this problem step-by-step in a logical way, using careful analytical reasoning to get the correct result. When you are done with your answer, provide your outputs in the format of **[your answer]**.
>
> ---
> We hope that our responses address your concerns in the review. Please let us know if you have any further questions!

---

> > ### Comment · Reviewer_uieZ · 2024-11-26
> >
> > Thanks for your replies, which answer my questions.  I will stay with my grade.

---

> > > ### Author Response · Authors · 2024-11-26
> > > **Response to Reviewer uieZ**
> > >
> > > Thank you for your reply, and we are glad that our responses cleared up the questions that you had! If our response has resolved your concerns, we would like to kindly ask you to consider raising the score for our work. Please let us know if you have any feedback on how we can further improve our submission, which we would be happy to address. Thank you very much again for your time!

---

> > > > ### Author Response · Authors · 2024-12-02
> > > > **Friendly Reminder: Discussion Period Deadline**
> > > >
> > > > Dear Reviewer uieZ,
> > > >
> > > > Thank you again for your reply, and for your valuable feedback on our initial version. As the discussion period will be ending shortly, we would be glad to address any remaining questions; if our responses and revisions have resolved your concerns, we would greatly appreciate it if you could consider improving the evaluation of our work. Thank you very much again for your time and consideration!

---

### Official Review · Reviewer_Entm · 2024-11-03

**Soundness:** 3
**Presentation:** 3
**Contribution:** 3
**Rating:** 6
**Confidence:** 4

**Summary:**

The paper explores LLMs' linguistic reasoning using linguistic puzzles on extremely low-resource languages. Its key contribution is a two-stage analogical prompting method, where the model first generates examples from related languages and then applies these to deduce grammar rules in a target language.

**Strengths:**

**Originality**

The paper introduces an innovative approach to evaluating linguistic reasoning in LLMs through analogical prompting. It applies this method to extremely low-resource languages and further evaluates generating exemplars through a different LLM, increasing overall performance.

**Quality**

The paper presents experimentation across multiple models and prompting strategies.

**Clarity**

The paper is well-structured, with clear explanations of each experimental setup, metric, and finding.

**Significance**

The paper highlights advancing the understanding of LLMs' reasoning capabilities across diverse languages. The focus on low-resource languages underscores the broader implications of this work for multilingual AI and low-resource language preservation.

**Weaknesses:**

1. Section 4 mentions that each response was manually evaluated to provide exact match scores, but this evaluation process lacks details. Specifically, there’s no mention of how many responses were reviewed, how many LLMs were involved, the number of evaluators, or their inter-annotator agreement. Without this, it’s challenging to assess the reliability of the manual evaluation.

2. Section 5.2 mentions other linguistic reasoning datasets, yet these were not utilized in the experiments. Incorporating additional benchmarks would provide more reliable and generalizable results.

**Questions:**

1. The paper briefly mentions that frontier models like GPT-4o and Llama-3.1-405B-Instruct often successfully identify language families. How accurately do LLMs identify language families, and how often do they correctly solve queries when the language family identification is accurate?

2. The results show that the mixture setting—where analogical exemplars are generated by one model and applied by another—outperforms the self-generation setting, but the paper does not delve deeply into why this occurs.

---

> ### Author Response · Authors · 2024-11-23
> **Response to Reviewer Entm**
>
> We thank Reviewer Entm for taking the time to review our paper, and for their positive review. We appreciate that the reviewer finds our contribution to be an “innovative approach” and our paper to be “well-structured”. Please see below for our responses to the reviewer’s concerns and questions.
>
> > Section 4 mentions that each response was manually evaluated to provide exact match scores, but this evaluation process lacks details. Specifically, there’s no mention of how many responses were reviewed, how many LLMs were involved, the number of evaluators, or their inter-annotator agreement. Without this, it’s challenging to assess the reliability of the manual evaluation.
>
>
> Evaluation is performed for exact match as the primary metric, due to the noted fallacies of ChrF2 and corpus-level BLEU scores in Section 4. All 272 problems were manually evaluated by one of the authors of this work; the annotation was purely for exact match (without partial scoring or other subjective notions for which inter-annotator agreement would be a useful signal). The sole necessity of human evaluation in using exact match is due to parsing errors in instruction following that we find with smaller models like Llama-3.1-8B and Aya-8B; to keep the evaluation protocol consistent this was repeated for all experiments.
>
>
> > Section 5.2 mentions other linguistic reasoning datasets, yet these were not utilized in the experiments. Incorporating additional benchmarks would provide more reliable and generalizable results.
>
> (Repeated from general response): The machine translation tasks -- more specifically, “Rosetta Stone” puzzles -- that form the primary focus of our work are present in a few benchmarks: PuzzLing Machines, modeLing, and LINGOLY. As we note in Section 5.2, modeLing was developed in part due to concerns of leakage of the problems in the PuzzLing Machines dataset, which is older (2020) and whose content may have been included in the vast web-scraping performed for curation of pre-training corpora. This has motivated our selection of the modeLing dataset, which consists entirely of newly written problems by experts to ensure the quality of the problems as well as avoid leakage. By contrast, the problems in LINGOLY are drawn from the UK Linguistics Olympiad, which may still be susceptible to leakage; the authors introduced a “no context” baseline, which is somewhat akin to our zero-shot baselines. However, the “no context” performance is non-zero for most mid-size and large / frontier models, inferred from their exact match and $\\Delta_{NC}$ scores, unlike in modeLing, suggesting that either leakage is present or the models are familiar with the languages being tested upon. This led us to rely on modeLing as our dataset of focus. Nonetheless, we appreciate your feedback on this matter; in the spirit of expanding the generalizability of our findings, we are currently working on evaluating our method on the LingOly dataset, which we will include in the camera-ready version.
>
> > The paper briefly mentions that frontier models like GPT-4o and Llama-3.1-405B-Instruct often successfully identify language families. How accurately do LLMs identify language families, and how often do they correctly solve queries when the language family identification is accurate?
>
> The claim we make regarding oracle vs inferred language families is more directly tied to frontier models not relying on language family labels, and leveraging an intrinsic understanding of language similarities to produce useful exemplars that result in performance gains. This is evident through the results with language isolates, wherein the model either chooses a language from a similar family higher in the taxonomy (often on the basis of geographical proximity) or assumes that the language is fictional or invented, and attempts to construct synthetic languages with similar syntactic patterns. That is, the model does not necessarily need to identify the correct language family in order to produce the correct answer. We include an analysis of the language families identified by GPT-4o and Llama-3.1-405B-Instruct in the inferred language families setting (as they often explicitly produce a label, e.g. "Ayutla Mixe belongs to the Mixe-Zoquean language family.”) in Appendix G.
>
> > The results show that the mixture setting—where analogical exemplars are generated by one model and applied by another—outperforms the self-generation setting, but the paper does not delve deeply into why this occurs.
>
> Without expert annotations, it is challenging to compare the exemplars generated by the different models, beyond their downstream impact on performance when applied with different deducers. As such, we can surmise from our results that a better deducer model (Llama-3.1-405B-instruct) can make the most out of qualitatively different exemplars.
>
> ---
> We hope that this addresses the points raised in the review, and we would be happy to address any additional questions!

---

> > ### Comment · Reviewer_Entm · 2024-11-26
> >
> > Thank you for the rebuttal. My concerns are addressed, but I still feel the paper needs some more work and insights (perhaps with the help of an expert) to identify and potentially mitigate issues that are currently present. However, I believe the paper does give insights into an area that is interesting and can be valuable to the community. I will keep my score.

---

> > > ### Author Response · Authors · 2024-11-26
> > > **Response to Reviewer Entm**
> > >
> > > Thank you very much for the reply! We greatly appreciate the feedback and your appreciation for our contributions.
> > >
> > > We would like to make a brief note on the challenge in introducing an expert — the extremely low-resource / nearly-extinct nature of the languages involved in these puzzles (e.g. Guugu Yimithirr only has an estimated 800 native speakers and a small population of scholars worldwide), and the typological rarities that may be present makes it quite difficult for a single expert to suffice. For instance, while an analysis of the grammar rules inductively learned for each language would be ideal (that is, the process guiding the source to target mappings formed), reliable verifiers for this do not exist, nor do we have grammar books from which to form a more grounded understanding of said rules. Thus, verifying the correctness (both for partial scoring in the target languages, as well as studying the correctness of analogical exemplars) could perhaps require a global crowd-sourcing initiative of many experts, which is quite challenging from a feasibility standpoint; we see this as beyond the scope of our work at present.

---

> > > > ### Comment · Reviewer_Entm · 2024-11-27
> > > >
> > > > I like the paper and agree that experts may not be available for extremely low-resource languages. But perhaps, olympiad winners from which the datasets are sourced can shed some more insights into how LLMs solve a problem versus how they approach it.
> > > >
> > > > I acknowledge the fact that expert feedback is not possible during the rebuttal period, but I feel like without that, the paper just feels like a collection of results without analysis beyond the correctness or not.

---

### Author Response · Authors · 2024-11-23
**General Response (Part 2/2)**

3. *Discussion of 2-Stage Analogical Prompting with Inferred Language Families*: To examine in further detail what happens during 2-stage analogical prompting (inferring the language families, without specifying that family through an oracle label, as in Figure 2b), we analyze the language family labels produced by Llama-3.1-405B-Instruct and GPT-4o in the first stage of our analogical prompting procedure. Note that this is prior to the model identifying specific languages from said family and producing the analogical exemplars. We compare these labels against the oracle labels in the table included in Appendix F, to yield a correctness score; we include these tables in Appendix G of our revised paper. Llama-3.1-405B-Instruct's language family correctness out of the 272 samples, relative to the oracle labels in Appendix F is $\frac{249}{272} = 91.54\%$, while GPT-4o's rate is $\frac{202}{272} = 74.26\%$. This reinforces our belief in the Llama models being strong multilingual reasoners. However, the model does not necessarily need to identify the correct language family in order to produce the correct answer. For instance, the Aya-35B exemplars applied in the weak-to-strong setting do not include any explicit family labels, jumping immediately into choosing similar languages and generating exemplars, which proves effective, as exhibited in Table 2.

### Revisions to Paper
We have posted a revised draft of our paper, incorporating the valuable feedback of the reviewers. We list the changes made below, which may also be visible in green in the revision:
* Added pointers in Sections 3.1 and 3.2 to their corresponding experiments in Sections 4.1 (baselines) and 4.2 (analogical prompting), respectively.
* Rephrased the final paragraph of Section 4.2.
* Updated Figure 1 to better illustrate the analogical prompting method and improve its clarity.
* Added a subsection (2.1) to discuss the linguistic reasoning (“Rosetta Stone”) puzzles studied in the work, with an example.
* Analyzed the language families identified in the self-generated analogical examplars with inferred families experiments and included this breakdown in Appendix G.

---

### Author Response · Authors · 2024-11-23
**General Response (Part 1/2)**

We sincerely thank the reviewers for their valuable feedback, comments, suggestions, and questions. We would like to clarify certain aspects of our work that were raised by the reviewers, and which we have sought to address in the revised version of our paper.

1. *Evaluation Method*: Our evaluation is primarily performed using exact match, given the stated concerns over corpus-level BLEU and ChrF2 scores (included nonetheless in the appendix), despite being a fairly strict criterion. This aligns with the scoring procedures for many computational linguistics competitions / linguistics olympiads. The introduction of a human annotator (an author of this work) to assess the generated response against the gold response is solely required to handle parsing issues in instruction following that arise with smaller models such as Llama-3.1-8B-Instruct and Aya-8B, to confirm that models are not being unfairly penalized despite producing the correct answer, but not in the desired boxed format as in the instruction. To ensure that the evaluation protocol was standardized across the board, this was repeated for all experiments, although stronger models (e.g. GPT-4o, Llama-3.1-405B-Instruct) were very adept at instruction following; hence, the exact match scores by parsing from the boxed responses and by human verification were the same for all experiments with those models as the generator. To reiterate, no human annotators were introduced to solve the problems on the Olympiad to verify their correctness, or determine where the model went wrong if the final answer was incorrect, as this would require an extremely experienced expert, as we note in our limitations section.

2. *Linguistic reasoning problems analyzed in our work*: The machine translation tasks -- more specifically, “Rosetta Stone” puzzles -- that form the primary focus of our work are present in a few benchmarks: PuzzLing Machines, modeLing, and LINGOLY. As we note in Section 5.2, the authors of modeLing suggest that this work was developed in part due to concerns of leakage of the problems in the PuzzLing Machines dataset, which is older (2020) and whose content may have been included in the vast web-scraping performed for the curation of pre-training corpora. This has motivated our selection of the modeLing dataset, which consists entirely of newly written problems by experts to ensure the quality of the problems as well as avoid leakage. By contrast, the problems in LINGOLY are drawn from the UK Linguistics Olympiad (UKLO), which may still be susceptible to leakage; to this effect, the authors introduced a “no context” baseline, which is somewhat akin to our zero-shot baselines. However, the “no context” performance is seemingly non-zero for most mid-size and large / frontier models, as inferred by their exact match and $\\Delta_{NC}$ scores, unlike in modeLing, suggesting that either leakage is present, or the models are extensively familiar with the languages being tested upon. This led us to rely on modeLing as our dataset of focus. Nonetheless, we appreciate the reviewers’ feedback on this matter, and with the spirit of expanding the generalizability of our findings, we are currently working on evaluating our method on the LingOly dataset, which we will include in the camera-ready version.

---

### Comment · Area_Chair_vB4e · 2024-11-25
**Action Required: Respond to Author Rebuttals - Nov 27**

Dear ICLR Reviewers,

The author discussion phase is ending soon. Please promptly review and respond to author rebuttals for your assigned papers. Your engagement is critical for the decision-making process.

Deadlines:
- November 26: Last day for reviewers to ask questions to authors.
- November 27: Last day for authors to respond to reviewers.
- November 28 - December 10: Reviewer and area chair discussion phase.

Thank you for your timely attention to this matter.

---

### Author Response · Authors · 2024-11-28
**Update: Paper Revision**

Dear Reviewers,

Thank you very much again for your valuable feedback, which has been helpful in improving our submission. Based on Reviewers SDWx and Entm's points raised on the generalizability of our findings, we have further evaluated our two-stage analogical prompting approach on the LINGOLY dataset [1] as well. The results reinforce the efficacy of our method, with remarkable improvements over the baseline and in fact, outperforming the state-of-the-art reported in their paper, to the best of our knowledge.

We include the results tables with GPT-4o below (comparing against the baseline with the same model in the LINGOLY paper), which are also reflected in both tabular and pictographic forms in Appendix D of our revised paper; these rely on the exact match metric. Note that the "breakthrough" level is the easiest set of problems and "Round 2" is the most difficult, which serves as the UK invitational qualification exam for the IOL. The columns represent the different question types present, which expand beyond the Rosetta Stone puzzle setting we explored with modeLing. Please note that all empty cells correspond to combinations for which problems do not currently exist in the LINGOLY dataset.

Baseline Results for GPT-4o (reported in LINGOLY):

|              | Computational | Text | Monolingual | Match-up | Pattern | Rosetta |
| :------------: | :-------------: | :----: | :-----------: | :--------: | :-------: | :-------: |
| Breakthrough |               | 100% |             |          | 47%     | 79%     |
| Foundation   | 0%            |      |             | 100%     | 67%     | 62%     |
| Intermediate |               |      |             |          | 58%     | 34%     |
| Advanced     |               |      | 0%          | 33%      | 53%     | 26%     |
| Round 2      |               |      | 0%          | 30%      | 27%     | 12%     |


Our Results with GPT-4o, 2-stage analogical prompting:

|              | Computational | Text | Monolingual | Match-up | Pattern | Rosetta |
| :------------: | :-------------: | :----: | :-----------: | :--------: | :-------: | :-------: |
| Breakthrough |               | 100% |             |          | 80%     | 86%     |
| Foundation   | 0%            |      |             | 100%     | 69%     | 80%     |
| Intermediate |               |      |             |          | 83%     | 64%     |
| Advanced     |               |      | 19%         | 50%      | 73%     | 51%     |
| Round 2      |               |      | 14%         | 42%      | 49%     | 41%     |


Deltas (Difference between our result and baseline):

|              | Computational | Text | Monolingual | Match-up | Pattern | Rosetta |
| :------------: | :-------------: | :----: | :-----------: | :--------: | :-------: | :-------: |
| Breakthrough |               | 0%   |             |          | +33%    | +7%     |
| Foundation   | 0%            |      |             | 0%       | +2%     | +18%    |
| Intermediate |               |      |             |          | +25%    | +30%    |
| Advanced     |               |      | +19%        | +17%     | +20%    | +25%    |
| Round 2      |               |      | +14%        | +12%     | +22%    | +29%    |


Encouragingly, we find that our results significantly outperform the baseline by a sizable amount across all difficulty levels, and across all tasks. Moreover, the results outperform the Claude-3 Opus state-of-the-art scores reported in the LINGOLY paper on every single setting, with the exception of the Breakthrough Rosetta Stone (easiest problems). Specifically, we find that our 2-stage analogical prompting method enables GPT-4o to solve questions of the monolingual type which it could not before (0% —> 19% and 14%); furthermore, the correctness rates jump considerably for some of the hardest categories over the baseline (1.81x improvement in Round 2 Pattern, 1.96x in Advanced Rosetta Stone, and 3.42x in Round 2 Rosetta Stone). It is especially worth noting that the Round 2 Rosetta Stone results corroborate with our findings on modeLing as reported in our paper. **These findings suggest that our method generalizes across both datasets and question types.**


### **Paper Revisions**

We have made the following revisions to our paper, to make reference to these results:
* The results have been added to Appendix D, with both tables and bubble plot-style visuals to highlight the performance gains across tasks and difficulty levels.
* We have added a note in Section 3.3 of the question types included in LINGOLY, which expand beyond the "Rosetta Stone" translation problem style.

[1] Bean, A. M., Hellsten, S., Mayne, H., Magomere, J., Chi, E. A., Chi, R., Hale, S. A., & Kirk, H. R. (2024). LINGOLY: A Benchmark of Olympiad-Level Linguistic Reasoning Puzzles in Low-Resource and Extinct Languages. arXiv preprint arXiv:2406.06196.

---

### Meta-Review · Area_Chair_vB4e · 2024-12-21

**Metareview:**

The paper investigates LLMs' linguistic reasoning capabilities through a novel two-stage analogical prompting method applied to low-resource language puzzles. The approach first generates examples from related languages and then uses these to deduce grammar rules in target languages.

Some reviewers acknowledge the strengths of the work, including comprehensive experimentation across multiple models and prompting strategies, clear presentation, and potential significance for multilingual AI and low-resource language preservation. The results show notable improvements over baselines. However, reviewers raise several concerns: 1) the paper's contribution is primarily empirical, with limited conceptual innovation beyond augmenting prompts with self-generated information; 2) there is a disconnect between the empirical results and the claimed conclusions about grammar rule learning, with insufficient analysis of the specific grammatical phenomena being tested; 3), the evaluation is limited to machine translation tasks, overlooking other important linguistic puzzle formats, and the results on cross-lingual transfer, while promising, do not provide substantial novel insights beyond existing work. While the authors have addressed some concerns through additional explanations and planned evaluations on other benchmarks, the questions about theoretical novelty and broader applicability remain (reviews from Reviewer KXR4 is not included in the consideration since no response from the reviewer).

Given these limitations, I agree with most reviewers suggesting the work falls marginally below the acceptance threshold of ICLR.

**Additional Comments On Reviewer Discussion:**

See above.

---

### Decision · Program_Chairs · 2025-01-22

Reject